# The Crystal Ball Hypothesis in Diffusion Models: Anticipating Object Positions from Initial Noise

**Yuanhao Ban[1], Ruochen Wang[1], Tianyi Zhou[2], Boqing Gong[3], Cho-Jui Hsieh[1], Minhao Cheng[4]**

[1]Department of Computer Science, University of California, Los Angeles
[2]Department of Computer Science, University of Maryland
[3]Google Research
[4]College of Information Sciences and Technology, Pennsylvania State University
`banyh2000, ruocwang@g.ucla.edu, chohsieh@cs.ucla.edu`
`tianyi@umd.edu, bgong@google.com, mmc7149@psu.edu`

## Abstract

Diffusion models have achieved remarkable success in text-to-image generation tasks, yet the influence of initial noise remains largely unexplored. In this study, we identify specific regions within the initial noise image, termed trigger patches, that play a key role in inducing object generation in the resulting images. Notably, these patches are **universal** and can be generalized across various positions, seeds, and prompts. To be specific, extracting these patches from one noise and injecting them into another noise leads to object generation in targeted areas. To identify the trigger patches even before the image has been generated, just like consulting the crystal ball to foresee fate, we first create a dataset consisting of Gaussian noises labeled with bounding boxes corresponding to the objects appearing in the generated images and **train a detector that identifies these patches from the initial noise.** To explain the formation of these patches, we reveal that they are **outliers** in Gaussian noise, and follow distinct distributions through two-sample tests. These outliers can take effect when injected into different noises and generalize well across different settings. Finally, we find the misalignment between prompts and the trigger patch patterns can result in unsuccessful image generations. To overcome it, we propose a reject-sampling strategy to obtain optimal noise, aiming to improve prompt adherence and positional diversity in image generation.

## 1 Introduction

In recent years, diffusion models have revolutionized the field of text-to-image generation (Saharia et al., 2022; Rombach et al., 2022; Dhariwal & Nichol, 2021; Nichol et al., 2021; Ho & Salimans, 2022). However, these models often fail to accurately adhere to the prompts, frequently generating objects with specific positions or attributes regardless of the input text (Chefer et al., 2023; Hertz et al., 2022; Wang et al., 2022). Despite various methods aimed at enhancing control over the generation process have been introduced, including modifying the denoising process (Balaji et al., 2022), manipulating cross-attention layers (Hertz et al., 2022; Feng et al., 2022), and retraining models using layout-image pairs (Zheng et al., 2023; Zhang et al., 2023; Voynov et al., 2023). Nonetheless, a key question remains unanswered: *Why is the generation process so difficult to control?*

In this paper, we show that Gaussian noise in the diffusion process plays a crucial role in image generation. Specifically, we discover the existence of **trigger patches** – distinct patches in the noise space that trigger the generation of objects in the diffusion model. By moving the trigger patch to a different position, the corresponding object will likely move to that location. Furthermore, this effect also exists across various prompts—the same trigger patch can initiate the generation of different objects, depending on the given prompt. These phenomena are illustrated in Figure 1: when replacing the target patch within **another initial noise** with the trigger patch, the injection position would generate an object. Identifying the location of trigger patches can provide insights into where objects will be generated **without running the diffusion process.** Moreover, moving/removing trigger patches can achieve certain image editing effects.

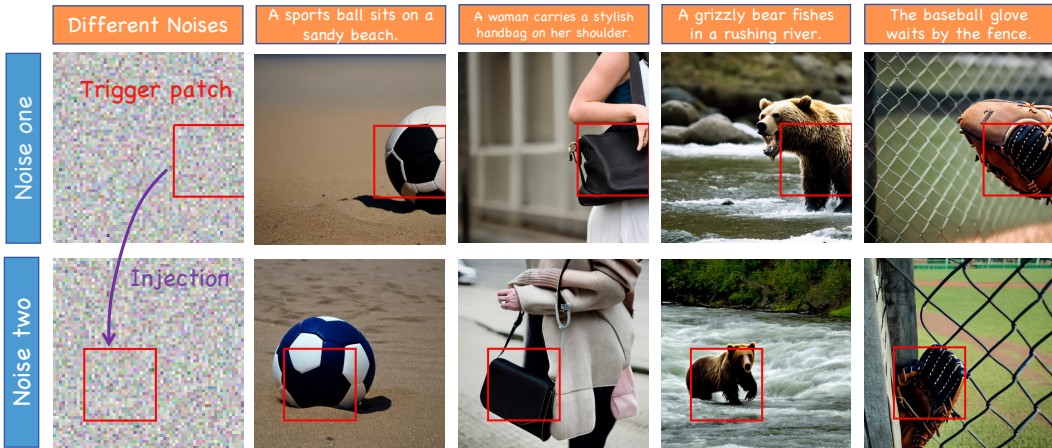

Figure 1: The first row displays images generated from the same seed, where the "trigger patch" can be identified within the red box, indicating its tendency to induce object generation. When this trigger patch is injected into another noise pattern, objects will appear in the corresponding position within the images generated from the mixed noise. Please refer to Fig 15 for more examples.

*How can we find trigger patches?* Our ultimate goal is to obtain a **Crystal Ball** that can anticipate the position of the objects in the generated images even before the diffusion process. We first propose a posterior approach based on the following intuition: given a specific initial noise, if all the objects are confined to a specific location regardless of the prompts, there must be a trigger patch in the corresponding position. Hence, we try to identify trigger patches by calculating the variance of coordinates of detected objects in the generated images. *But can we detect trigger patches without actually running a diffusion process?* To achieve that, we propose a "trigger patch detector", which functions similarly to an object detector but operates in the noise space. Our trigger patch detector, functioning like a crystal ball, achieves an mAP50 of 0.333 on the validation set[1], revealing that trigger patches are ignorant of specific prompts and noises.

*So what makes trigger patches special?* We hypothesize that the trigger patches are **outliers** within the Gaussian noise. We perform a two-sample test comparing the trigger patches to randomly selected noise patches and confirm that they follow distinct distributions. In particular, more effective trigger patches show greater divergence from the standard Gaussian distribution. To further support our claim, we design some handcrafted trigger patches shifted from the original distribution. We validate that they effectively trigger object formation at corresponding positions in the generated images.

Finally, we demonstrate two applications of trigger patches. For scenarios when location information is specified in the prompt, we show that designing noises with trigger patches at the target location can significantly boost the generation success rate from 57.08% to 83.64%. Conversely, for applications when location diversity of generated objects is preferred, we demonstrate that using our proposed detector to "purify" the noise can significantly increase generation diversity.

## 2 TRIGGER PATCHES DETERMINE GENERATED LOCATIONS

### 2.1 PRELIMIARIES

**Denoising Diffusion Probabilistic Models (DDPM)** DDPM (Ho et al., 2020) is a kind of generative model, which achieves great performance on high-quality image synthesis. The inference process initiates with the standard Gaussian noise $x_T$ and iteratively applies the model to progressively denoise it back towards the natural image $x_0$ that subjects to the real data distribution.

**Classifer-free guidance for conditional generation** Text-to-image diffusion models incorporate classifier-free context information into the reverse diffusion process via cross-attention layers. Specifically, at each sampling step, the denoising error is calculated by adjusting the conditional error with an unconditional error, factored by a guidance strength.

---

[1]Faster R-CNN Ren et al. (2015) achieves a $mAP_{50}$ of 0.596 on COCO validation set Lin et al. (2014).

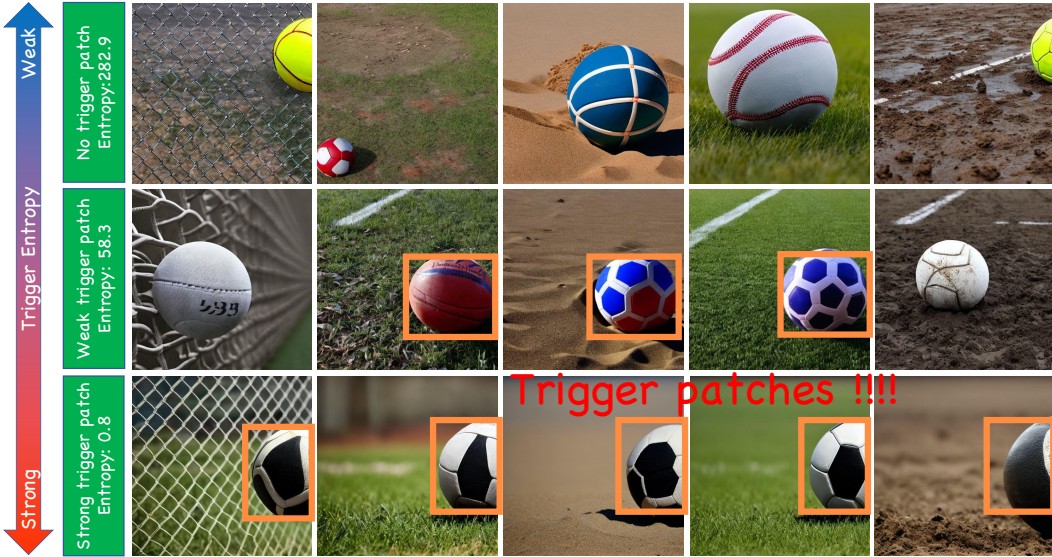

Figure 3: Illustration: Each row shows five images generated from one seed. The patch in the bottom row has the strongest effectiveness in inducing the generation of the object, while the objects in the top row are dispersed. So there must exist a strong trigger patch in the orange bounding box in the bottom row noise.

## 2.2 DEFINITION OF TRIGGER PATCH

Where will a diffusion model position an object in the generated images? In this section, we show that *the initial noise takes an important role in generated objects' location*. Specifically, we identify regions within the initial noise, termed "**trigger patches**", that largely determine the object location. Formally, a trigger patch is a patch in the noise space with the following properties: (1) Triggering Effect: When it presents in the initial noise $Z_0$, the trigger patch consistently induces object generation at its corresponding location; (2) Universality Across Prompts: The same trigger patch can trigger the generation of various objects, depending on the given prompt.

Specifically, for a fixed initial noise, we generate 25 images with various prompts, and then detect the object in each generated image. Please refer to Appendix B for more details. By summing the object masks and normalizing the results, we generate a heatmap where each pixel represents the probability of an object appearing at that location. Fig 2 demonstrates the heatmaps for two different noises. For the image on the left, all generated objects are concentrated in roughly the same location with high probability, indicating the presence of a trigger patch that consistently drives object generation at that location. Conversely, the right heatmap has no such an obvious pattern, suggesting no trigger patch in that particular noise.

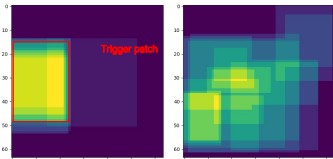

Figure 2: Heatmaps of generated objects on two different noises. The left one has a trigger patch while the right one does not. Please refer to Fig 8 for more visualized results.

To formulate a quantitative expression, we can first use a pre-trained detector to obtain the coordinates of the bounding boxes $\{(x_1, y_1, x_2, y_2)\}$, where $(x_1, y_1)$ represents the coordinates of the upper-left corner, and $(x_2, y_2)$ represents the coordinates of the lower-right corner of the bounding box. Then, we can define a posterior metric about the effectiveness of the regions in inducing objects, termed "trigger entropy". For each initial noise, the more concentrated the boxes are, the smaller trigger entropy it has. To avoid the bias of the object size, we compute the center point of each object bounding box $x_c = \frac{x_1 + x_2}{2}$ and $y_c = \frac{y_1 + y_2}{2}$. Then the entropy of the patch can be defined based on the variance of the box centers:

$$\mathcal{H}(X) = \frac{1}{2}\left(\frac{1}{n}\sum_{i=1}^{n}(x_{c_i} - \bar{x}_c)^2 + \frac{1}{n}\sum_{i=1}^{n}(y_{c_i} - \bar{y}_c)^2\right) \tag{1}$$

where $\bar{x}_c = \frac{1}{n}\sum_{i=1}^{n} x_{c_i}$, $\bar{y}_c = \frac{1}{n}\sum_{i=1}^{n} y_{c_i}$, and $i$ is the box index. So, the more entropy a region has, the less effective it is in determining the objects' locations, as shown in Fig 3.

## 3 THE CRYSTAL BALL: DETECTING TRIGGER PATCHES FROM NOISE

### 3.1 TRAINING A CRYSTAL-BALL DETECTOR

The above method requires generating images to identify trigger patches. *Can we detect the trigger patches without running the diffusion process, just like looking into the Crystal Ball to anticipate it?* To answer this question, we try to train a trigger patch detector, which functions similarly to an object detector but operates in the noise space.

**Generating Noise-Annotations Dataset** To train a detector, we generate a dataset consisting of initial noises paired with the bounding boxes of objects in the corresponding images. Initially, we select five classes from the COCO dataset: "stop sign, bear, sports ball, handbag, and apple". We then prompt ChatGPT to generate five unique sentences for each class, ensuring each sentence features only one object to prevent duplicates. Please refer to Appendix B for the prompt used for ChatGPT and the 25 prompts used to generate images. We sample 20,000 Gaussian noises and use diffusion models to generate images based on these noises and prompts. A pre-trained object detector from MMDetection (Chen et al., 2019) is then applied to identify the object bounding boxes, which are resized from the image space to the latent space. Only the bounding box with the highest score and correct object label is retained for each image. Please refer to Appendix B for more details about image generation settings and the detection settings.

**Detection Dataset** We divide 20,000 random noises into a training set (17,500), a validation set (1,000), and a test set (1,500). Subsequently, we modify the trigger patch annotations to create four distinct datasets, the details of which are presented in Table 1. The *Augmented* dataset uses bounding boxes of all the 25 prompts, while the *Restricted* only uses those of the class "sports ball". In the *Class-Specific* dataset, the model is required to output the class labels of the trigger patch. Additionally, we craft a *Shuffled* dataset with shuffled annotations to serve as a baseline to mitigate the effects of other bias. For example, if we use an biased training dataset with all the objects appearing in one place, a detector simply outputs center boxes constantly can achieve good results. We call it model and prompt bias. So we propose to shuffled the annotations of the data, as Fig 9 shows. If all bounding boxes are centralized, the training outcomes on the Shuffled dataset should align with those from the Restricted dataset.

Table 1: Dataset Description

| Dataset | Classes | Annotations per noise | Output classes | Shuffled annotations |
|---|---|---|---|---|
| Restricted | Sports ball | $5 = 1 \times 5$ | × | × |
| Augmented | All | $25 = 5 \times 5$ | × | × |
| Class-Specific | All | $25 = 5 \times 5$ | ✓ | × |
| Shuffled | Sports ball | $5 = 1 \times 5$ | × | ✓ |

**Results** We utilize the MMDetection repository to train our trigger patch detector. Please refer to Appendix D for more details about training recipe. The results can be seen in Table 2. **Our detector on Restricted achieves the mAP$_{50}$ of 0.325, surpassing the Shuffled baseline by 0.124.** Such evidence eliminates the influence of the model bias and prompt bias, verifying that the model has learned to locate the trigger patches from initial noises. Please refer to the Appendix D for more comparison with detectors on standard COCO dataset. Notably, the results on the Augmented dataset show that the trigger patches can generalize across different text prompts and provide a foundation for the applications of our detector in various scenarios. Meanwhile, the detector on Class-Specific dataset shows degenerated performance, indicating that the trigger patches are ignorant of the classes and universal across prompts. In other words, these patches can only determine where objects are generated, not what to generate.

Table 2: Training a detector to anticipate object positions from initial noise.

| Dataset | Restricted | Augmented | Class-Specific | Randomized |
|---------|-----------|-----------|----------------|------------|
| $\text{mAP}_{50}$ | 0.325 | 0.333 | 0.091 | 0.201 |

## 3.2 THE UBIQUITOUS AND DIVERSITY OF TRIGGER PATCH

In this subsection, we perform a series of studies to address the following questions: **1) Dataset statistics:** How frequently do trigger patches appear in the initial noise? **2) Generalization:** Can trigger patches generalize effectively across various noises, sampling configurations (schedulers, sampling steps, and aspect ratios) and different model types (LoRA models, fine-tuned models, and others)? **3) Trigger-prompt interaction:** How does the interaction between the trigger patch and prompt affect performance, especially when the prompt includes positional information that contradicts or aligns with the trigger patch location? **4) Multiple trigger patches:** What happens when a noise contains multiple trigger patches? **5) Trigger patch preference:** Do trigger patches show a preference for certain object categories?

**Dataset Statistics** We compute the trigger entropy defined in equation 1 for all the 20,000 noises and plot the histograms in Fig 4. We first study a trivial case, where we compute the trigger entropy based on the boxes of one class, "sports ball", and five prompts about this class. Then, for a more general one, we compute the trigger entropy based on all 5 classes and 25 prompts to see if the metric can generalize well across various prompts and classes. As shown in Fig 4, nearly **10%** of the noises in the first group have an entropy near 0. In other words, for nearly 2,000 noises, sports balls always show up in almost the same place across the five images generated from an initial noise, indicating that strong trigger patches are very common in initial noises.

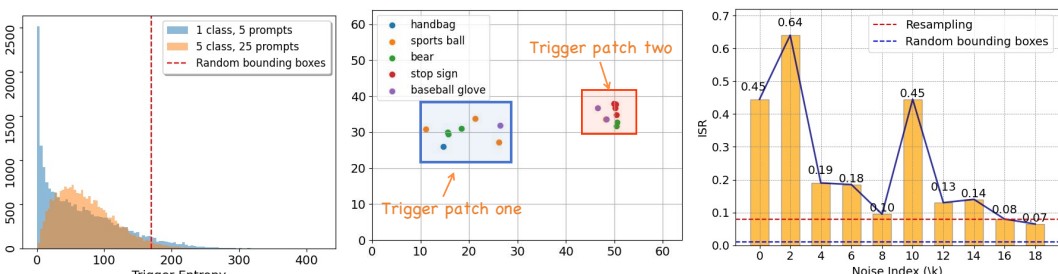

Figure 4: Trigger Entropy Distribution on the created dataset. Randomly selected bounding boxes as a baseline.

Figure 5: Object Distribution: For one noise, we plot the scatter to see where the center of the bounding boxes are dispersed.

Figure 6: Trigger Injection: We sort the 20,000 trigger patches by Trigger entropy and do injection experiments on every 2000 noise.

**Generalization** We design an **Trigger Injection** experiment to test if these patches can still take effect when injected into another noise. In particular, we replace an arbitrary patch in a noise map with the trigger patch and then use the blended noise for generation. A successful injection is characterized by the detected bounding box occupying more than 75 percent of the trigger patch region. Then we define the injection success rate (ISR) as the ratio of successful injection cases to the total number of cases. We sort the patches in our dataset by trigger entropy and select one every two thousand for this test. We do the injection experiment with 200 new random noises. There are two baselines as follows. 1) Resampling: Resample Gaussian noise for the target region maintaining the same mean and variance. 2) Random: Select a random patch within a source noise, which might overlap with the trigger patch in the source noise. Figure 6 illustrates that the Injection Success Rates (ISRs) of most trigger patches exceed those of the Resampling (0.08) and Random (0.01) baselines, confirming the efficacy of the trigger patches. Additionally, a lower trigger entropy correlates with a higher ISR, suggesting that the trigger patches with low entropy are particularly effective in determining object locations. Moreover, to verify the trigger patches are "universal" and can **generalize well across different sampling timesteps, samplers, aspect ratios, fine-tuned models and LoRA models**, we conduct a series of additional Trigger Injection experiments with different configurations and models. Please refer to the Appendix C for more ablation study on generalization.

**Trigger-Prompt Interaction** The ability of trigger patches to introduce positional information raises an important question: What happens when prompts also include positional cues, and how do the prompt and noise interact when their object positions are **either aligned or in conflict**? To explore this, we first design prompts with the format of "a coco class name on the right side". Then we select three noises: 1) Aligned: with a strong trigger patch on the right, 2) Contradicted: with a strong trigger patch on the left, and 3) Dispersed: with trigger patches dispersed throughout the image. After viewing the generated images, we divided them into four groups: 1) Aligned: The position of the object is aligned with the prompt guidance (On the right side). 2) Contradicted: The position of the object contradicts with the prompt guidance (On the left side). 3) Duplicated: the generated image has two objects on both sides. Please refer to Fig 14 for more details. 4) Hard to judge (Occupying the entire picture, failed generation, or in the middle of the image). Please to Appendix H for a failed analysis. Table 3 demonstrates that the trigger patches significantly influence the final positions

Table 3: Diversity Results

| Noise | Aligned (%) | Contradicted (%) | Duplicated (%) | Hard to judge (%) |
|---|---|---|---|---|
| Aligned | 63.5 | 6.3 | 6.3 | 23.9 |
| Contradicted | 35.0 | 32.5 | 8.8 | 23.7 |
| Dispersed | 25.0 | 10.0 | 11.3 | 53.7 |

of objects. When the prompt and trigger patches are contradictory, 32.5% of objects adhere to the trigger patches, while only 35.0% follow the prompt's guidance, highlighting the prevalence of failed generations due to **conflicts between the prompt and initial noise about where to position objects.** Notably, when the prompt and trigger patches are aligned, 63.5% of objects are accurately positioned. These results indicate the potential to manipulate trigger patches for controlled image generation, as demonstrated in Section 5.2.

**Multiple trigger patches within a noise** It contains two questions. *1) Can multiple trigger patches appear in one noise?* To answer the question, we plot the center points of the detected bounding boxes of one specific noise in Fig 5. As shown, two distinct clusters are present, indicating the existence of multiple trigger patches within a single noise. *2) Can noises with multiple trigger patches benefit multiple object generation performance?* We conduct a series of experiments using the prompts like "a lion and a yellow clock". **We find the presence of multiple trigger patches enhances multiple object generation performance.** Please refer to the Appendix G for more details.

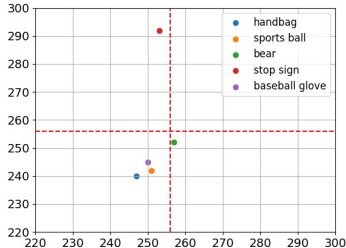

Figure 7: We plot the averaged center points for the bounding boxes of five classes. As the image size is 512×512, we marked the centered baseline of 256 with a red dashed line. The variance of these central points is greater on the vertical axis.

**Trigger Patch preference** The trigger patches themselves have no significant preference for certain kinds of objects and does not depend on the specific prompt — both trigger patches in Fig 5 can generate baseball gloves (purple points) and bears (green points). However, we do observe some indications of positional preference, as certain objects are more likely to generate within the trigger patches located at specific positions, revealing a position bias of these object classes. As shown in Fig 5, handbags only appeared in the bottom left region, while stop signs appeared only in the upper right region. **We attribute this preference to the position of the trigger patch within the initial noise, rather than the properties of the patch itself.** In the real-world distribution, most stop signs are located in the upper part of the image, while most handbags are in the lower part. This likely causes them to favor trigger patches in certain positions. To verify it, we analyze the statistics of the generated dataset to see the preferred positions of the classes. We calculate the averaged center points for the bounding boxes of five classes across over 20000*25 images and plot these averaged center points in Fig 7. As shown, stop signs tend to appear in higher positions, while handbags favor lower positions. Another finding is that these objects do not exhibit a positional bias horizontally, clustering near the 256 dashed baseline.

## 4 WHAT CONTRIBUTES TO THE FORMATION OF TRIGGER PATCHES

In this part, we propose a potential explanation for the existence of trigger patches. While previous work Guo et al. (2024); Sun et al. (2024); Zheng et al. (2023) mainly focuses on the cross-attention

maps and analyzes the interactions between specific noise, prompts, and models, we hypothesize that **these trigger patches are typical outliers in the sampled noise**, and have minimal correlation with prompts. Furthermore, we propose a method to synthesize trigger patches with great success.

## 4.1 TWO-SAMPLE TEST

An energy-based two-sample test measures the distance between the distributions of two samples by computing the sum of pairwise Euclidean distances among all sample points (Székely & Rizzo, 2013). Meanwhile, the test includes $p$-value to statistically quantify the extent to which the distributions's difference. A low $p$-value (typically less than 0.05) suggests that the two sets of samples are not subjected to the same distribution. Please refer to the Appendix E for detailed mechanism and the python code we use for the 2-sample test. To construct a trigger patch group, we select 2,000 noises, compute the center point of each bounding box, and extract $24{\times}24$ trigger patches around the center. We then create a negative group by randomly selecting 2,000 patches of $24{\times}24$ in the same noises. Additionally, we randomly select another group of patches of the same size to serve as a control group. **The results show that the $p$-value between the trigger patch group and the negative groups are 0.0, while the $p$-value between the negative group and the control group is 0.938.** Hence, the trigger patches and those random patches in the negative groups are from different distributions. Furthermore, we also find a clear negative correlation between the trigger entropy and the energy distance. Please refer to the Appendix E.3 for more details.

## 4.2 HAND-CRAFTED TRIGGER PATCHES

If the trigger patches are indeed outliers, it may be feasible to construct certain noises that deviate from a normal Gaussian distribution to serve as artificially created trigger patches. To this end, we conduct experiments to test various artificial trigger patches.

- Natural trigger patches: We extract the trigger patches with the lowest trigger entropy from the 20,000 noise samples.
- Resampling baseline: We resample Gaussian noise maintaining the same mean and variance to establish a baseline.
- Random baseline: We randomly select 25 sets, each consisting of four numbers, from a range of 512. These numbers are interpreted as the coordinates for the final detected bounding boxes, upon which we compute entropy. This baseline is designed to simulate scenarios where each patch has an equal probability of generating an object.
- Shifted Gaussian: We sample Gaussian noises with altered standard deviation (std) so the resultant noise follows a different Gaussian from the diffusion models' default noises.
- Sine Function: We create sinusoidal noise patches by firstly using a function that applies a sine transformation to each coordinate axis $(x, y, z)$ and then summing up the values. We then add them to an initial noise via interpolation as follows,

$$\overline{P_{(x,y,z)}} = \sin(\theta) \cdot \left[ \sin(\frac{2\pi x}{l_x}) + \sin(\frac{2\pi y}{l_y}) + \sin(\frac{2\pi z}{l_z}) \right] + \cos(\theta) \cdot P_{(x,y,z)}, \quad (2)$$

  where $P_{(x,y,z)}$ is the pixel value of the original patch at position $(x, y, z)$, $l_x$, $l_y$ and $l_z$ are the widths of the trigger patch, and $\theta$ is the interpolation parameter. In our case, $l_x$ $l_y$ and $l_z$ are 24, 24 and 4, respectively.

The results are displayed in Table 4. Natural trigger patches achieved a success rate of 44.5%, which exceeds the rates of randomly selected bounding boxes and the resampling of the region by 42.5% and 36.0%, respectively. Notably, Sin Function patches yield comparable results when moderate interpolation parameters are set, achieving an ISR of 49%. When the interpolation parameter $\theta$ is set at $0.15{\cdot}\frac{\pi}{2}$, it achieves a significantly high ISR of 81.0%. However, this setting also causes image distortion. Meanwhile, sampling Gaussian noise with a larger standard deviation appears to be more effective in inducing trigger patches than using a smaller one, verifying that these trigger patches might be considered outliers.

## 5 APPLICATIONS

The ability to detect trigger patches enhances control over object locations in generated images, opening the door to numerous applications. Here, we demonstrate two such applications: 1) In

Table 4: Main results on the injection experiments. The higher the ISR, the more effective the patch. The left major column shows the ISRs for the baseline methods: Random, Resampling, and Natural. The middle major column shows the results for the shifted Gaussian method with different standard deviations(STD). The right major column shows the results for the Sine Function with different interpolation weights.

| Baselines | | Shifted Gaussian | | Sine Function | |
|---|---|---|---|---|---|
| | ISR(%) | STD | ISR(%) | $\theta(\cdot\frac{\pi}{2})$ | ISR(%) |
| Random | 1.0 | 0.8 | 8.5 | 0.08 | 33.5 |
| Resampling | 8.5 | 1.2 | 29.0 | 0.10 | 49.0 |
| Natural | 44.5 | 1.5 | 90.0 | 0.15 | 81.0 |

scenarios where the prompt does not have positional information, we aim to increase the positional diversity of the generated images. 2) Conversely, when the prompt includes explicit positional guidance, our goal is to ensure that the generated images follow the provided directions. It is important to note that manipulating trigger patches alone may not deliver state-of-the-art results. Our main objective is to demonstrate that even such a straightforward approach can achieve reasonable performance, highlighting the significant role that trigger patches play in the generation process. Note that we use the detector obtained in Sec 3 to identify and manipulate trigger patches in both applications, rather than relying on off-the-shelf trigger patches obtained through posterior methods. **This demonstrates the open-vocabulary capability of our approach.**

### 5.1 ENHANCED LOCATION DIVERSITY BY REMOVING TRIGGER PATCHES

**Background**   Researchers have observed various biases in diffusion models, including gender bias (Luccioni et al., 2024) and color bias (Orgad et al., 2023). However, position bias, which is the tendency for objects to consistently appear in the same locations across different generations, has received minimal attention. This type of bias has significant implications, particularly in synthetic data generation using diffusion models. For instance, when generating an automobile dataset, a model exhibiting position bias might consistently place roadblocks in the left-bottom corner of the image, leading to skewed data. Moreover, such a bias can affect the diversity of generated images in applications like ChatGPT, where similar images are generated for varied prompts due to the presence of trigger patches in the initial noise.

**Method**   To mitigate this, we have developed a method involving the use of the detector for reject sampling. This process begins with the detection of trigger patches in the initial noise. If the confidence scores of the bounding boxes exceed a predefined threshold, the region within the box is flagged for regeneration. Our approach is designed to ensure that the noise used for generation is "pure" and has no strong trigger patches that could confine object placement. Please refer to Fig 11 for more visualized illustrations.

**Baselines**   Although researchers have studied the positional bias in GAN-based generative models (Choi et al., 2021), no counterpart works have been found on the diffusion models. Here are the baselines: Initno: focusing on the role of initial noise, Initno (Guo et al., 2024) may be the most related one, though it needs the specific prompt for each generation process to take effect. Control: we establish a Control group by generating images using standard methods without reject-sampling. Random: we select Random bounding boxes of size 24×24 from the initial noise. This approach simulates a scenario where all pixels have an equal probability of generating an object, which is our target. Attend (Chefer et al., 2023): we utilize the *StableDiffusionAttendAndExcitePipeline* class from diffusers (von Platen et al., 2022) repository. We select the token IDs corresponding to the most relevant words. For example, for the prompt "a ball on the left", we configure the token IDs as "2,5", which correspond to the words "ball" and "left". This ensures the model focuses on these specific elements during image generation. Attention Refocusing Phung et al. (2024): we use the code from the official repository. We use GPT-4 to generate layout bounding boxes. For each image, we prompt the model to generate a new layout. Structured (Feng et al., 2022): we use the code from the official repository and apply the default configurations in the README for the demos.

**Metrics and Configurations**   We calculate the trigger entropy for the bounding boxes in these groups to assess the impact of our method on diversity and positional bias. The entropy is calculated

as outlined in equation 1, with higher values indicating greater diversity. **The higher the Entropy is, the better.** Our experiment setup includes generating images from 10 random prompts described by ChatGPT, followed by image generation using diffusion models. The objects' bounding boxes are identified using the same COCO detector in Sec 3, and entropy calculations are conducted on the bounding boxes with confidence scores over 0.75. We set the rejection threshold for the confidence score of the detector as 0.6 and conducted our experiments using 1000 different seeds. Note that the prompts we use are in an open-vocabulary setting and include many objects beyond the five classes used to train the detector. Please refer to the Appendix F.1 for the prompts we use and experiment configurations.

Table 5: Entropy values for different methods in location diversity.

| Methods | Control | Initno | Attend | Refocusing | Structured | Random | Ours |
|---|---|---|---|---|---|---|---|
| Entropy | 135.97 | 139.89 | 145.16 | 102.097 | 133.92 | 170.64 | **171.84** |

As we can see from the table, images generated by our methods show great diversity in position, with an entropy of 171.84, surpassing the control group by 31.95, very close to the pure random outputs, which is 170.64. Meanwhile, Initno shows bad performance with an entropy of 139.89, which is behind ours by a large margin. Other baselines also perform bad, which may result from the fact that they are designed for setting a specific layout, not enhancing diversity. Please refer to the Appendix F.1 for visualized results.

## 5.2 BETTER PROMPT FOLLOWING WITH TRIGGER PATCHES

**Background and Methods**   Research indicates that diffusion models often struggle to adhere to the positional information specified in prompts. This issue is typically due to the appearance of trigger patches in unintended locations as shown in the trigger-prompt interaction experiment in Sec 3.2. For instance, a prominent trigger patch on the left side of an image can conflict with a prompt like "a dog on the left", resulting in generation failures. To address this, we propose a reject-sampling technique to obtain an optimal initial noise where trigger patches align with the prompt requirements even before generation. Specifically, we continue to resample noises until the center point of the bounding box, which scores highest according to our detector, is positioned within the area targeted by the prompt.

**Prompts**   To assess our method's effectiveness, we design 10 prompts incorporating the words **right** or **left** and generate 500 images for each prompt. Please refer to Appendix F.2 for more details about the prompts we used.

**Metrics**   We generated images using 10 prompts that incorporate the words "right" or "left", eg, "a sports ball in the left". We then used a pre-trained COCO detector from MMDetection to verify whether the objects were correctly positioned according to the prompt. Supposing the x-axis coordinates of the left and the right edges of the bounding boxes are $x_1$ and $x_2$ respectively, and the image has a size of 512×512, we define *the generated object in the left part of the image* as $\frac{x_1+x_2}{2} < \frac{512}{2}$. If *the generated object in the left part of the image* and the prompt specifies "left", we consider it a successful case. We calculate the GSR as the ratio of the number of successful cases to the total number of cases. The higher the GSR is, the more effective the method.

**Results**   As seen in Table 6, our method, relying solely on the initial noise, achieves an impressive guidance success rate (GSR) of 83.64%. While it is slightly surpassed by Attention Refocusing, it offers substantial advantages in terms of simplicity and efficiency. **Our approach requires no knowledge of the model or prompts and does not rely on access to a large language model.** Moreover, using an NVIDIA RTX A6000, our method generates an image in approximately **5** seconds, whereas Attention Refocusing takes around **15** seconds. Our method introduces a novel approach to controllable generation by utilizing trigger patches even before the diffusion generation process, providing an alternative to the traditional practice of manipulating cross-attention maps through analysis. Please refer to Appendix F.2 for more visualization results.

Table 6: Prompt following Results

| Methods | Control | Random | Attend | Attention Refocusing | Structured | Ours |
|---|---|---|---|---|---|---|
| GSR(%) | 57.08 | 61.08 | 64.52 | 89.83 | 77.41 | **83.64** |

## 6 RELATED WORK

**Initial noise**   In DDIM (Song et al., 2020), the final image is determined by the model parameters, the text prompt, and the initial noise. While the model parameters and text prompts have been extensively researched, the initial noise remains relatively under-explored. Mao et al. (2023) first highlights that initial noises exhibit distinct preferences for certain layouts and analyze this phenomenon through the lens of cross-attention. Sun et al. (2024) utilized an inverted reference image with finite inversion steps to incorporate valuable spatial awareness, leading to a new framework for controllable image generation. Guo et al. (2024) attributed failed generation cases to "bad initial noise" and proposed optimizing the initial noise via cross-attention to achieve better results. On the other hand, Lin et al. (2024) claim that common schedulers do not have zero terminal SNR at the final timestep, which fails to reflect the reality that the model is given pure noise at inference, creating a gap between the training and inference stages. In contrast, our claim is that the initial pure noise is not entirely "pure" and contains specific trigger patches that can induce object generation during the inference stage.

Building on these studies, our research focuses on the role of initial noise. Unlike previous works that qualitatively suggest the presence of positional information in initial noise, we provide evidence for **the existence of trigger patches that tend to generate specific objects** by defining a metric to quantitatively assess the effectiveness of these patches. Based on the findings above, we managed to train a detector, which can detect trigger patches even before the image generation process. Additionally, we propose that these trigger patches are universal and capable of generalizing across different positions, prompts, and background noises. While prior research primarily examines noise from a cross-attention perspective, we discover that the intriguing properties might stem from **the statistical characteristics of these patches**, which are outliers within the initial noise. We identify significant applications in enhancing positional diversity. While previous studies mainly addressed biases related to gender, color, and texture, our findings reveal that noise with strong trigger patches can introduce positional bias. Finally, in applications, previous methods require performing diffusion steps and accessing cross-attention maps. In contrast, our approach does not rely on any knowledge of the prompt or the model. Simply applying the detector to the noise is sufficient to anticipate the position of the generated object.

**Types of objection detection**   Object detection is a fundamental computer vision technique that identifies and locates objects in digital images and videos. Various detection networks have been developed for different application scenarios, such as static images (Ren et al., 2015), video frames (Wojke et al., 2017), depth images (Reading et al., 2021), point clouds (Qi et al., 2017), sequential data (Lea et al., 2017), and multi-channel images (Chen et al., 2016). However, we propose a novel and counter-intuitive approach: using **pure Gaussian noise** as the input, the model is trained to learn from these noises and extract patches with intriguing properties. To the best of our knowledge, this is the first study to propose such an approach.

## 7 CONCLUSION

In this study, we have uncovered specific trigger patches within the initial noise that are likely to induce object generation. We subsequently developed a detector capable of identifying these trigger patches from the initial noise prior to the generation process. We characterized these trigger patches as outliers within Gaussian noise and conducted experiments to validate this classification. Finally, we show two potential applications for our detector, both of which have demonstrated excellent performance. We leave more applications for future work.

## 8 LIMITATIONS

One limitation is the size of the dataset we used for training the trigger patch detector. Due to limited computational resources, we only train on 20,000 noises and 500,000 images, so the performance of our detector can be potentially improved with a larger training set.

ACKNOWLEDGEMENTS

This research is supported by supported by NSF 2048280, 2325121, 2244760, 2331966 and ONR N00014-23-1-2300:P00001..

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

## A    VISUALIZATION

## B    DATASET CREATION

**Prompts** We use ChatGPT to generate sentences based on the given object name. The prompt is as follows:

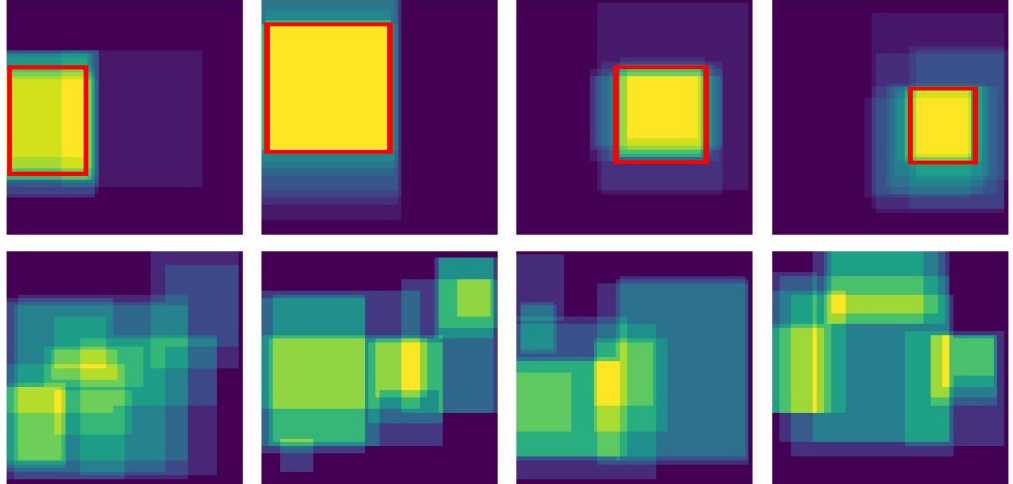

Figure 8: More examples to verify the existence of the trigger patch. We summing the object masks of the images generated from a single noise and normalize the result. Each pixel of the resulted heatmap represents the probability of an object appearing at that location. The ones in the upper row have a trigger patch in the red bounding box. While the bounding boxes in the bottom row are much more diverse, indicating there is no strong trigger patch in the area.

---

**Prompt**

## Task:
Generate a sentence describing a scene that includes the given object, ensuring no additional objects are mentioned.
## Example:
Input: sports ball
Output: A sports ball is caught in a fence.
## Task:
I will give you 5 kinds of objects, 'sports ball', 'baseball glove', 'bear', 'handbag', 'stop sign'. Please generate 5 sentences for each object.

---

**Generated Sentences**    The final prompts for the five objects are as follows:

---

**Sports Ball**

1. A sports ball is caught in a fence.
2. A sports ball lies forgotten under a tree.
3. A sports ball sits on a sandy beach.
4. A sports ball rests on a grassy land.
5. A sports ball stands out on a muddy field.

---

**Baseball Glove**

1. The baseball glove waits by the fence.
2. A young athlete breaks in a new baseball glove.
3. A baseball glove rests on the dugout bench.
4. His baseball glove hangs by the door.
5. A baseball glove is left on the fence during practice.

---

> **Bear**
>
> 1. A grizzly bear fishes in a rushing river.
> 2. A bear cub explores the forest with curiosity.
> 3. A bear catches a fish in the river.
> 4. A black bear forages for berries in the woods.
> 5. A bear sniffs the forest floor.

> **Handbag**
>
> 1. A fashionable handbag complements an elegant outfit.
> 2. A woman carries a stylish handbag on her shoulder.
> 3. A handbag rests on a cafe table during lunch.
> 4. A handbag holds essentials for a day of shopping.
> 5. A handbag adds a pop of color to a monochrome look.

> **Stop Sign**
>
> 1. A red stop sign halts traffic at an intersection.
> 2. A stop sign stands alone on a country road.
> 3. A stop sign is covered in a layer of snow.
> 4. A stop sign is adorned with event flyers.
> 5. A stop sign stands at the entrance to a neighborhood.

**Image generating**  We adopt the Stable Diffusion V2 Rombach et al. (2022) from diffusers von Platen et al. (2022) to generate images. On an NVIDIA 2080TI GPU, generating a single image takes approximately 12 seconds. In total, producing 500k images on an 8-GPU machine takes around 7 days.

**Detector**  We adopt RTMDet Lyu et al. (2022) from the MMDection repo Chen et al. (2019) for its efficiency and good accuracy. During inference, we observed that multiple bounding boxes are often proposed for a single image. For the annotations detected by the COCO detector, we retain only those with a confidence score above 0.75, and where the predicted class matches the given prompt.

## C  GENERALIZATION

In this section, we conduct a series of ablation experiments to verify the trigger patches are "universal" and can take effect across different settings. We verified the effectiveness of the trigger patches by following the protocol of the **Generalization paragraph** described in Sec 3.2. Specifically, we injected the trigger patches into different random noise samples and used the blended noise to generate images. We then determined the success of the injection by checking whether the targeted region generated the intended object. A successful injection is defined as a detected bounding box occupying more than 75 percent of the trigger patch region. The Injection Success Rate (ISR) is calculated as the ratio of successful injections to the total number of cases, with **a higher ISR indicating a more obvious trigger patch phenomenon**.

### C.1  GENERALIZATION RESULTS FOR DIFFERENT SAMPLING CONFIGURATIONS

In this subsection, we evaluate whether the trigger patch generalizes effectively across different sampling steps, aspect ratios, and schedulers.

We conduct experiments using Stable Diffusion v2 provided by Diffusers von Platen et al. (2022). The default configuration is set to 50 timesteps, a PNDM Liu et al. (2022a) scheduler, and an 1:1 aspect ratio. For each configuration, we also conducted a resampling baseline, where Gaussian noise was resampled within the target patch while maintaining the same mean and variance. **A significant gap between the experimental group and the baseline indicates the effectiveness of the trigger patch in those settings.** The higher the GAP is, the more effective it is. The results are as follows:

Table 7: Generalization Results for Different Time Steps.

| Time Steps | 30 | 40 | 50 | 60 |
|---|---|---|---|---|
| Resampling Baseline | 0.100 | 0.085 | 0.105 | 0.090 |
| Experimental Group | 0.645 | 0.665 | 0.660 | 0.630 |
| Gap | 0.545 | 0.580 | 0.555 | 0.540 |

Table 8: Generalization Results for Different Deterministic andStochastic Schedulers.

| Deterministic Schedulers | | | |
|---|---|---|---|
| Schedulers | Resampling Baseline | Experimental Group | Gap |
| PNDM Liu et al. (2022a) | 0.105 | 0.660 | 0.555 |
| LMSD Karras et al. (2022) | 0.095 | 0.620 | 0.525 |
| EulerDiscrete Karras et al. (2022) | 0.115 | 0.620 | 0.505 |
| HeunDiscrete Karras et al. (2022) | 0.110 | 0.610 | 0.500 |
| DDIM Song et al. (2020) | 0.145 | 0.630 | 0.485 |
| **Stochastic Schedulers** | | | |
| Schedulers | Resampling Baseline | Experimental Group | Gap |
| DPM Lu et al. (2022) | 0.180 | 0.690 | 0.510 |
| KDPM Karras et al. (2022) | 0.240 | 0.405 | 0.165 |
| EulerAncestral Karras et al. (2022) | 0.180 | 0.165 | -0.015 |
| TCDScheduler Zhou et al. (2024) | 0.105 | 0.650 | 0.545 |
| DDPM Ho et al. (2020) | 0.145 | 0.155 | 0.010 |

In the three tables, an obvious gap can be observed between the Resampling baseline and the Experiment Group, showing that our trigger patch can induce the generation of the object successfully across different sampling configurations. As observed from the Tab 8, our method underperforms with certain stochastic samplers. The reason lies in the nature of stochastic samplers, which introduce noise to the latent variables at each diffusion step. If the added noise is large enough, it can overwrite the initial noise to some extent, diminishing the influence of our trigger patch. Conversely, if the introduced noise is small, our trigger patch is still able to exert a strong effect. To test this hypothesis, we conducted a series of experiments using the DDIM Song et al. (2020) sampler from the diffusers, which includes a parameter eta to control the strength of the introduced noise: It has a parameter $eta$ which controls the strength of the introduced noise. Setting $eta = 0.0$ makes the sampler behave like a deterministic one. The suggested value for $eta$ is 0.1-0.3. In Tab 10, a negative correlation between eta and the ISR Gap is observed. This suggests that the noise level directly impacts the trigger patch's ability to influence the diffusion process.

## C.2 GENERALIZATION RESULTS OF DIFFERENT MODELS

In this subsection, we evaluate whether the trigger patch generalizes well across different models. We selected three types of models and adopted the same approach as described in the previous subsection.

1. **SOTA T2I models:** We selected Stable Diffusion XL Podell et al. (2023).

2. **Finetuned models:** We used the off-the-shelf model "oraul/finetuned_stable-diffusion-v1-4_FFHQ_smaller_ep_2" from the Hugging Face Hub.

3. **LoRA models:** Most LoRA models often overfit certain objects and show degraded performance. Therefore, we used style-LoRA models that focus on altering the style of the model output. Specifically, we employed two adapters from Hugging Face: Water Color (available at "ostris/watercolor_style_lora_sdxl") and Crayons (available at "ostris/crayon_style_lora_sdxl").

4. **DiT models:** We adopts Stable Diffusion v3 Esser et al. (2024) with a **DiT** backbone. To ensure compatibility, we modified the shape of the input latent to produce output images of the same size as those generated by other Stable Diffusion models. For implementation, we utilized the code from Hugging Face, using the model ID "stabilityai/stable-diffusion-3-

Table 9: Generalization Results for different Aspect Ratios.

| Aspect Ratios | 16:9 | 4:3 | 1:1 | 3:4 | 9:16 |
|---|---|---|---|---|---|
| Resampling Baseline | 0.030 | 0.090 | 0.105 | 0.050 | 0.030 |
| Experimental Group | 0.440 | 0.630 | 0.660 | 0.475 | 0.500 |
| Gap | 0.410 | 0.540 | 0.555 | 0.425 | 0.470 |

Table 10: Effect of eta Parameter in the DDIM Sampler on ISR Gap.

| eta | Resampling Baseline | Experimental Group | Gap |
|---|---|---|---|
| 0.0 | 0.145 | 0.630 | 0.485 |
| 0.01 | 0.140 | 0.620 | 0.480 |
| 0.05 | 0.180 | 0.635 | 0.455 |
| 0.1 | 0.165 | 0.640 | 0.475 |
| 0.2 | 0.145 | 0.555 | 0.410 |
| 0.5 | 0.160 | 0.355 | 0.195 |

medium-diffusers". This adaptation allowed us to integrate the unique architecture of the DiT backbone while maintaining consistency in output dimensions for fair comparisons.

Table 11: Generalization Results of Different Models

| Models | SDXL | Oraul | Lora Water Color | Lora Crayons | SD3 |
|---|---|---|---|---|---|
| Resampling Baseline | 0.355 | 0.330 | 0.265 | 0.305 | 0.219 |
| Experimental Group | 0.500 | 0.525 | 0.565 | 0.695 | 0.441 |
| Gap | 0.145 | 0.195 | 0.300 | 0.390 | 0.222 |

These results indicate that our method can generalize well across different models.

## D  DETECTOR TRAINING

**Shuffled Data**   We establish a **reasonable baseline** by permuting the annotations of different noises, which prevents the detector from receiving any meaningful information from the noise. Please refer to the Fig 9 for more details.

**Training**   To train a detector capable of anticipating trigger patches, we utilize the MMDetection repository Chen et al. (2019). Our approach integrates Generalized Focal Loss Li et al. (2020) and a ResNet101 with a $4\times$ width expansion He et al. (2016). We follow the COCO dataset configurations provided by the repository. To handle noise with dimensions of $4 \times 64 \times 64$, we set the input dimension to 4. During training, we observed that strong data augmentation techniques, such as Flip and Random Resize, significantly impaired performance, leading us to eliminate them. The training was conducted on four NVIDIA GTX 2080 Ti GPUs with 17.5k noise samples, and the loss converged after 20 epochs. We report the mean Average Precision (mAP50) threshold on the 1k validation dataset.

**Comparison with object detectors on COCO detector**   Our method achieves a mAP$_{50}$ of 0.333, while Faster R-CNN Ren et al. (2015) achieves a mAP$_{50}$ of 0.596 on COCO validation set Lin et al. (2014).

However, due to the complexity of our task, direct comparisons with state-of-the-art detectors on real-world images may not be reasonable. Here are the challenges for our task:

1. **Different Input Characteristics**: Our inputs are Gaussian noises, which differ significantly from real-world images that have clear boundaries and semantic segments. Thus, a direct comparison is meaningless. Meanwhile, we believe we are among the **first** to explore this task in such a context. There is still a long way to go for our settings.

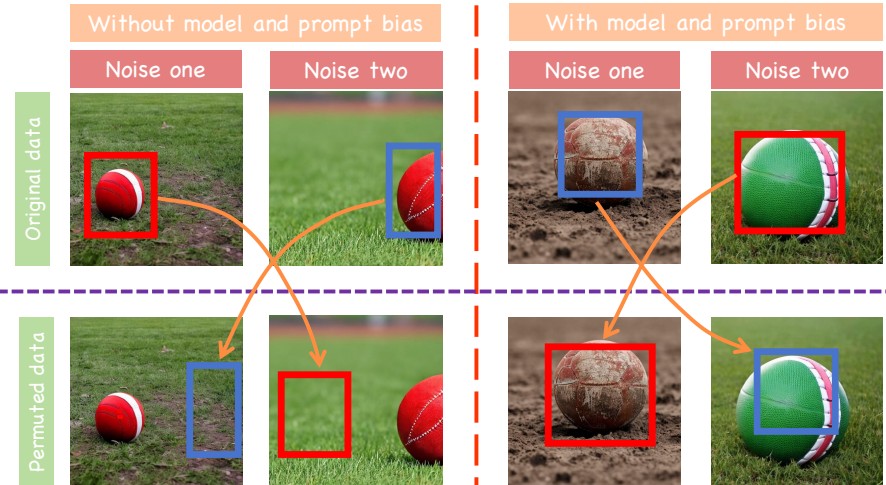

Figure 9: Shuffled Data: We shuffled the annotations associated with the noise samples. Initially, in the top row, Noise One was labeled with Annotation One, and Noise Two with Annotation Two. After permutation, shown in the bottom row, Noise One is now labeled with Annotation Two, and Noise Two with Annotation One. Ideally, in scenarios with a strong bias, permuting the annotations should have minimal or no impact on the final results. This is because, in both shuffled and unshuffled datasets, the bounding boxes remain centrally located within the image.

2. **Object Size Variation**: Our trigger patches are universal and **do not depend on specific prompts**. However, we observed that object sizes vary greatly. For instance, in a $512 \times 512$ image, the average area for a bear is 69,274 pixels, whereas for a sports ball, it is only 29,919 pixels. Despite this variation, our detector achieves comparable performance, demonstrating its capability to accurately detect the **center points** of trigger patches even under these challenging conditions.

3. **Challenges with Advanced Techniques**: We experimented with various configurations, finding that most advanced techniques, which typically improve object detection, did not benefit our task. Techniques like random cropping in data augmentation even reduced performance, possibly due to the nature of our input data. As a result, we simplified our training configuration to retain the statistical properties as much as possible.

Based on the reasons above, we establish a **reasonable baseline** by permuting the annotations of different noises, which prevents the detector from receiving any meaningful information from the noise. This baseline achieves an mAP50 of **0.201**, while our method achieves an mAP50 of **0.333**, clearly highlighting the capacity of our detector. Additionally, our applications in Sec 5.1 and Sec 5.2 are all based on the detector we trained. The **successful application of our detector in Section 4** further validates its effectiveness.

# E   ENGERY-BASED TWO-SAMPLE TEST

## E.1   THE MECHANISM

First, the energy distance quantifies the distance between two probability distributions and computes the energy distance between the two tested sets, named $T_{\text{observed}}$. Then, the two samples are combined into one joint sample, which is randomly shuffled multiple times (e.g., 1000 times). After each random permutation, the shuffled sample is split into two new samples. For each random permutation, the statistic for the newly formed samples, denoted as $T_{\text{permuted}}$, is computed. The p-value for the permutation test is calculated as:

$$\text{p-value} = \frac{\text{Number of times } T_{\text{perm}} \geq T_{\text{obs}}}{\text{Total number of permutations}}$$

In our experiments, we find it extremely time-consuming to compute the energy distance, so we set the sampling time to 1000. In this case, the p-value can be 0.0 if all $T_{\text{perm}}$ values are smaller than $T_{\text{observed}}$. We believe that 1000 permutations are sufficient to ensure the p-value remains below 0.001, which is considerably smaller than the conventional threshold of 0.05.

## E.2 PYTHON CODE

Below is the Python code we used for the two-sample test:

```python
import numpy as np
from scipy.spatial.distance import cdist
from tqdm import tqdm

def energy_distance(x, y):
    """Compute the energy distance between two datasets"""
    x, y = np.asarray(x), np.asarray(y)
    xy_distance = np.mean(cdist(x, y, 'euclidean'))
    xx_distance = np.mean(cdist(x, x, 'euclidean'))
    yy_distance = np.mean(cdist(y, y, 'euclidean'))
    return 2 * xy_distance - xx_distance - yy_distance

def permutation_test(x, y, num_permutations=1000):
    """Perform a permutation test to compute the p-value
    for the energy distance"""
    combined = np.vstack([x, y])
    original_distance = energy_distance(x, y)
    count = 0

    for _ in tqdm(range(num_permutations)):
        np.random.shuffle(combined)  # Randomly shuffle the data
        new_x = combined[:len(x)]
        new_y = combined[len(x):]
        permuted_distance = energy_distance(new_x, new_y)
        if permuted_distance >= original_distance:
            count += 1

    p_value = count / num_permutations
    return original_distance, p_value
```

Listing 1: Two Sample Test Code

## E.3 A NEGATIVE CORRELATION BETWEEN THE TRIGGER ENTROPY AND THE DISTANCE ENERGY

To study the correlation between the trigger entropy and the distance energy, we divide 20,000 noises into ten groups according to the order of Trigger Entropy with 2,000 noises each and the first group has the lowest Trigger Entropy. Then we compute the center point of each bounding box, and extract a 24×24 trigger patch around the center. Following these steps, we obtain a list of trigger patches for each group. We then create ten negative groups by randomly selecting a patch of 24×24 in the same noise. Additionally, we craft a control group by randomly selecting patches of the same size. **The results show that all the $p$-values between trigger patch groups and negative groups are 0.0, while the $p$-value between the negative prompts and the control group is 0.938.** Note that a $p$-value of 0.0 in our settings actually means that the true $p$-value is under 0.001, detailed in the Appendix E. Hence, the trigger patches and those random patches in the negative groups are from different distributions. Furthermore, we examine the relationship between the energy distance of trigger patches from negative patches and the trigger entropy, as illustrated in Figure 10. This figure reveals a clear negative correlation: the greater the deviation of a patch from the original distribution, the more likely it is to be a trigger patch characterized by low trigger entropy. This observation again supports our hypothesis that these trigger patches are outliers within the noise.

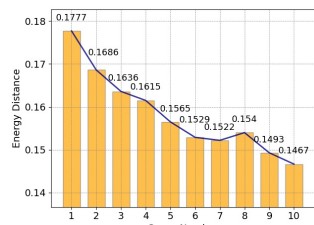

Figure 10: Energy Distance

# F APPLICATIONS

## F.1 ENHANCED LOCATION DIVERSITY BY REMOVING TRIGGER PATCHES

**Prompts** We prompt ChatGPT to generate sentences describing complex scenes to test the **open-vocabulary generation of our method**, besides the 5 coco classes. The prompts we use are as follows:

> **Prompts for Location diversity**
>
> 1. The golden sunlight filters through the dense canopy of the forest, casting dappled shadows on the moss-covered ground.
> 2. A red bicycle leans against a gnarled oak tree, its wheels slightly caked with mud from the morning's ride.
> 3. Nearby, a picnic table is set with a checkered cloth, and atop it rests a basket filled with fresh fruit and sandwiches.
> 4. A frisbee lies forgotten on the grass, a few feet away from a sleeping dog with its fur glistening in the sun.
> 5. In the background, a kite dances in the sky, its bright colors a stark contrast against the blue expanse above.
> 6. A laptop is open on the table, displaying vibrant images of nature, momentarily abandoned for the allure of the outdoors.
> 7. A baseball glove and ball sit on the bench, remnants of a game played in the spirit of friendly competition.
> 8. A traffic cone marks the end of a nearby trail, signaling caution to the cyclists and hikers passing by.
> 9. A traffic cone marks the end of a nearby trail, signaling caution to the cyclists and hikers passing by.
> 10. As the day wanes, the street lights begin to flicker on, their glow adding a soft luminescence to the tranquil scene.

**Experiment settings** For each prompt and each method, we generate 1000 images. Please refer to Fig 11 for visualization results. We adopt an off-the-shelf detector Lyu et al. (2022) from the MMDection repo Chen et al. (2019) to detect the bounding box and calculate the average entropy. The higher the average entropy is, the better.

## F.2 BETTER PROMPT FOLLOWING APPLICATION BY INJECTING TRIGGER PATCHES

**Prompts** We designed 10 prompts with direct positional information to test the methods.

> **Prompts Targeted at Left**
>
> 1. a sports ball in the left.
> 2. a cow in the left.
> 3. an apple in the left.
> 4. a bicycle in the left.
> 5. a vase in the left.

> **Prompts Targeted at Right**
>
> 1. a sports ball in the right.
> 2. a cow in the right.
> 3. an apple in the right.
> 4. a bicycle in the right.
> 5. a vase in the right.

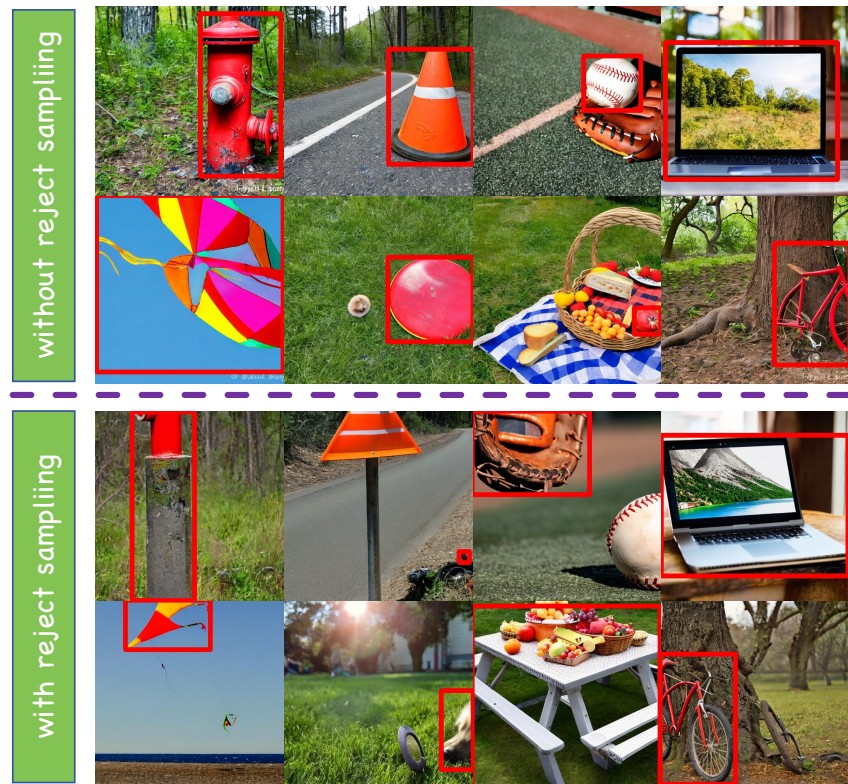

Figure 11: We visualize the generated images in the location diversity application in Sec 5.1. The upper part shows the results generated by one noise without rejection sampling, and the objects tend to cluster to the right edge of the picture. Then we generate a noise with no obvious trigger patch with the help of the detector obtained in Sec 3 and the reject sampling technique. The bottom part shows the results generated by that enhanced noise, which shows great diversity in object location.

**Visualized results** Please refer to the Fig 12 for visualized results.

## G    MULTIPLE OBJECTS

We conducted additional experiments to assess our method's ability to handle multiple objects. Following the protocol outlined in Attend-and-Excite Chefer et al. (2023), we used the same prompts provided in their GitHub repository and generated 64 images for each prompt. Below are some examples of the prompts used:

> **Testing prompts**
>
> 1. a lion and a yellow clock.
> 2. a bird and a black bowl.
> 3. a monkey and a red car.
> 4. a mouse with a bow.
> 5. a frog and a purple balloon.

For our method, we place two separate trigger patches on the left and right sides of the image. This allows us to prevent the two objects from fusing into each other. Specifically, the latent has a size of $64 \times 64$. We place the two trigger patches within the bounding boxes defined by the coordinates (0, 20, 24, 44) and (40, 64, 24, 44), where the four coordinates represent the top-left and bottom-right corners of each bounding box in the format $(x_1, x_2, y_1, y_2)$. This ensures that the patches are placed in distinct regions of the image, preventing any overlap or fusion of the objects associated with these triggers.

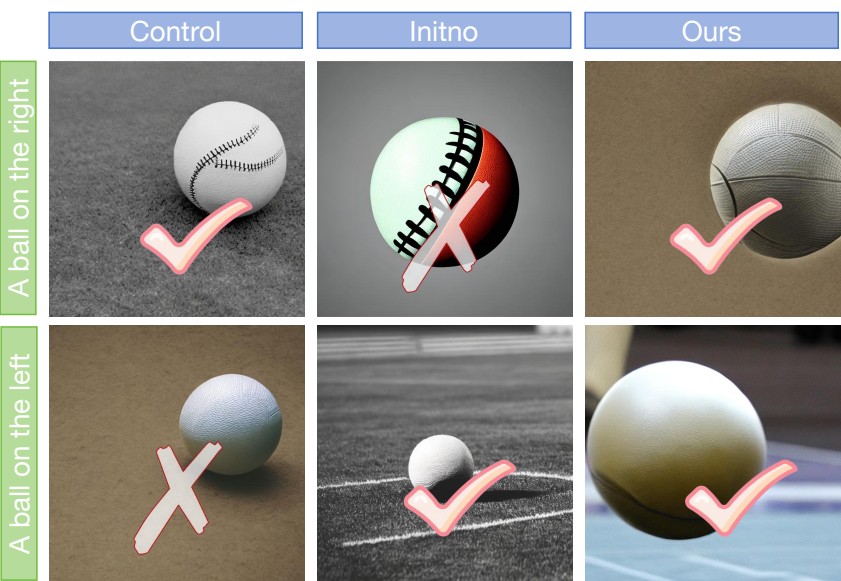

Figure 12: Visualization results of better prompt following application by injecting trigger patches in Sec 5.2. A check mark and a cross indicate whether the generated image is correct or not, respectively.

We adopt Composable Diffusion Liu et al. (2022b), Structure Diffusion Feng et al. (2022), Attend-and-Excite Chefer et al. (2023) and Divide-and-Bind Li et al. (2023) as our baselines. Please refer to Fig 13 for more visualized results.

We then evaluated the results using three metrics following the protocol in Attend-and-Excite Chefer et al. (2023): CLIP Full Image-Text Similarity (CLIP-Full), CLIP Minimum Image-Text Similarity (CLIP-Min), and BLIP-CLIP Text-Text Similarity (BLIP-Text), as proposed in the paper. We also utilize a human-preferenced model, PickScore Kirstain et al. (2023) by comparing the results of these methods against the control baseline. The score reflects the model's preference for the images generated by each method relative to the baseline. For all these metrics, **higher values indicate better performance**.

The results are as follows:

Table 12: Comparison of CLIP and BLIP-Text metrics for different methods.

| Metrics | CLIP-Full | CLIP-Min | BLIP-Text | PickScore |
|---|---|---|---|---|
| Stable Diffusion | 0.341 | 0.251 | 0.792 | 0.500 |
| Composable Diffusion | 0.348 | 0.252 | 0.768 | 0.514 |
| Structure Diffusion | 0.349 | 0.246 | 0.780 | 0.527 |
| Attend-and-Excite | 0.352 | 0.263 | **0.830** | 0.514 |
| Divide-and-Bind | 0.349 | 0.261 | 0.822 | 0.534 |
| Ours | **0.360** | **0.264** | 0.826 | **0.557** |

In the CLIP-Full, CLIP-Min and Pickscore metrics, our method achieved the best results, surpassing the strongest baselines designed for this task by 0.008 and 0.001, respectively. Additionally, our method does **not require any intervention in the attention maps** during the generation process, significantly reducing the computational resources needed and making it both model and prompt agnostic.

## H    FAILED ANALYSIS FOR THE TRIGGER-PROMPT INTERACTION.

There are mainly three reasons for the failed cases of aligned experiments.

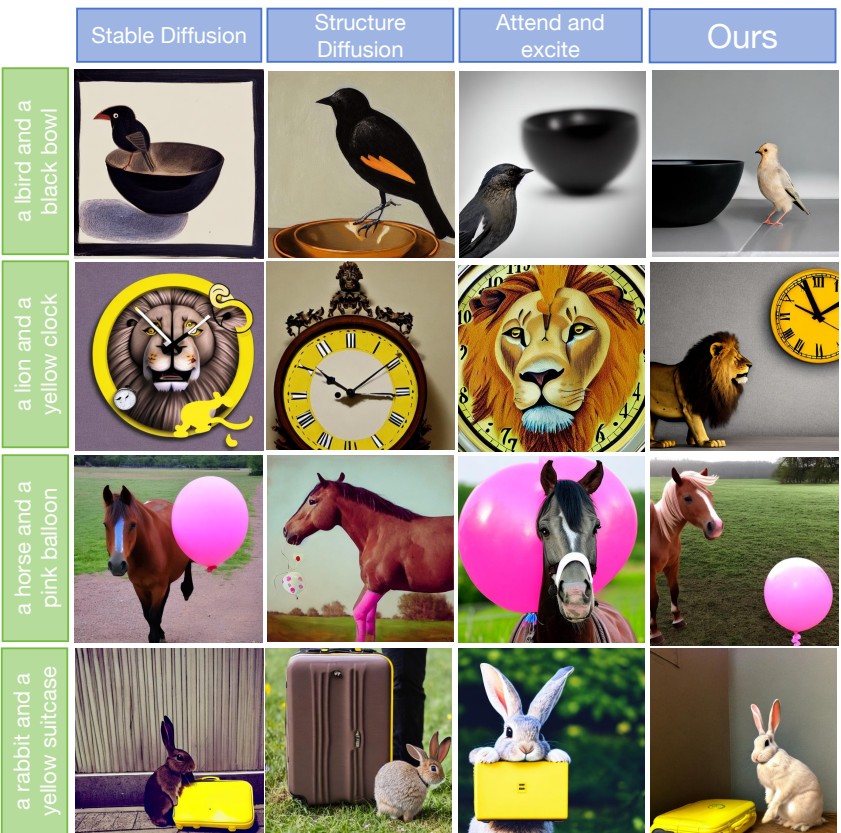

Figure 13: Visualized results of multiple objects generation.

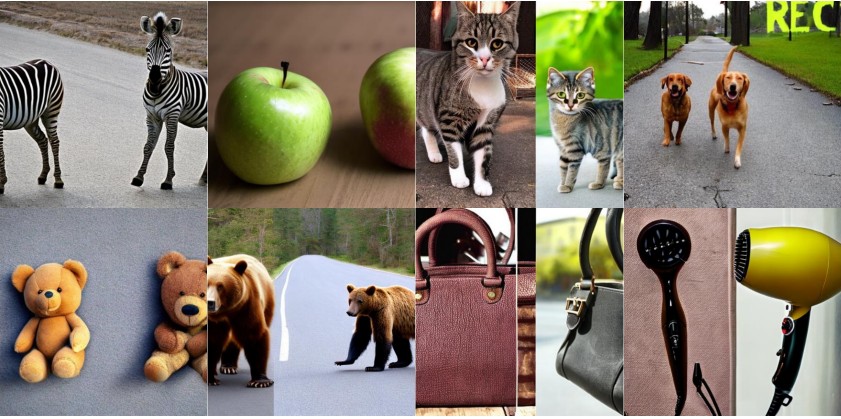

Figure 14: Visualized results of the duplicate phenomenon.

Firstly, since our trigger patches are universal, any object can be generated at the designated position. Occasionally, irrelevant objects may occupy this position. For instance, given the prompt "a stop sign in the street" with the trigger patch located in the right corner, the position might instead be occupied by a tree. Please refer to Fig 16 for the visualized results.

Secondly, some classes we use have difficulty determining whether they are on the left or right. For example, for the class "fork," the model often generates a long fork that stretches across the image horizontally.

Finally, in some cases, the objects are placed almost in the middle of the image and It is hard to judge if it is correct. These two cases will be categorized into "hard to judge".

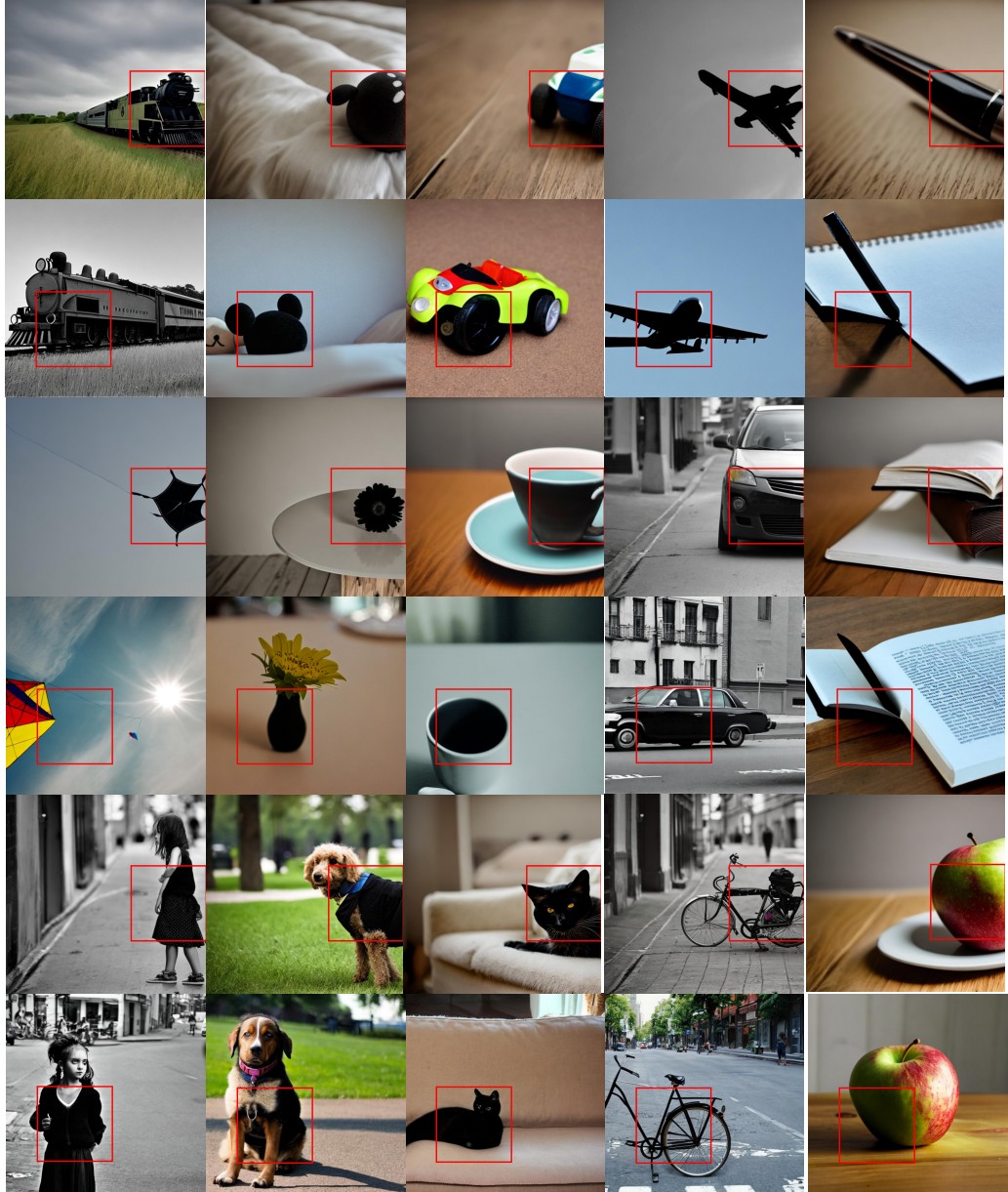

Figure 15: More examples to verify the generalization ability of the trigger patch for Fig 1. The first, third, and fifth rows display images generated from the same noise pattern, while the second, fourth, and sixth rows are derived from a blended noise pattern. Each pair of adjacent rows shares the same prompt. Within the red box, the "trigger patch" can be observed, showcasing its tendency to influence object generation. When this trigger patch is incorporated into the blended noise pattern, objects emerge in the corresponding positions in the images generated from the mixed noise.

We can observe that although "aligned" only has 63.5% , most of the rest samples belong to "hard to judge", and there are very few samples belonging to "contradicted" and "duplicated". In fact, if we ignore the hard-to-judge cases, there are 83.4% cases belonging to "aligned" while only 16.6% cases belong to "contradicted" and "duplicated".

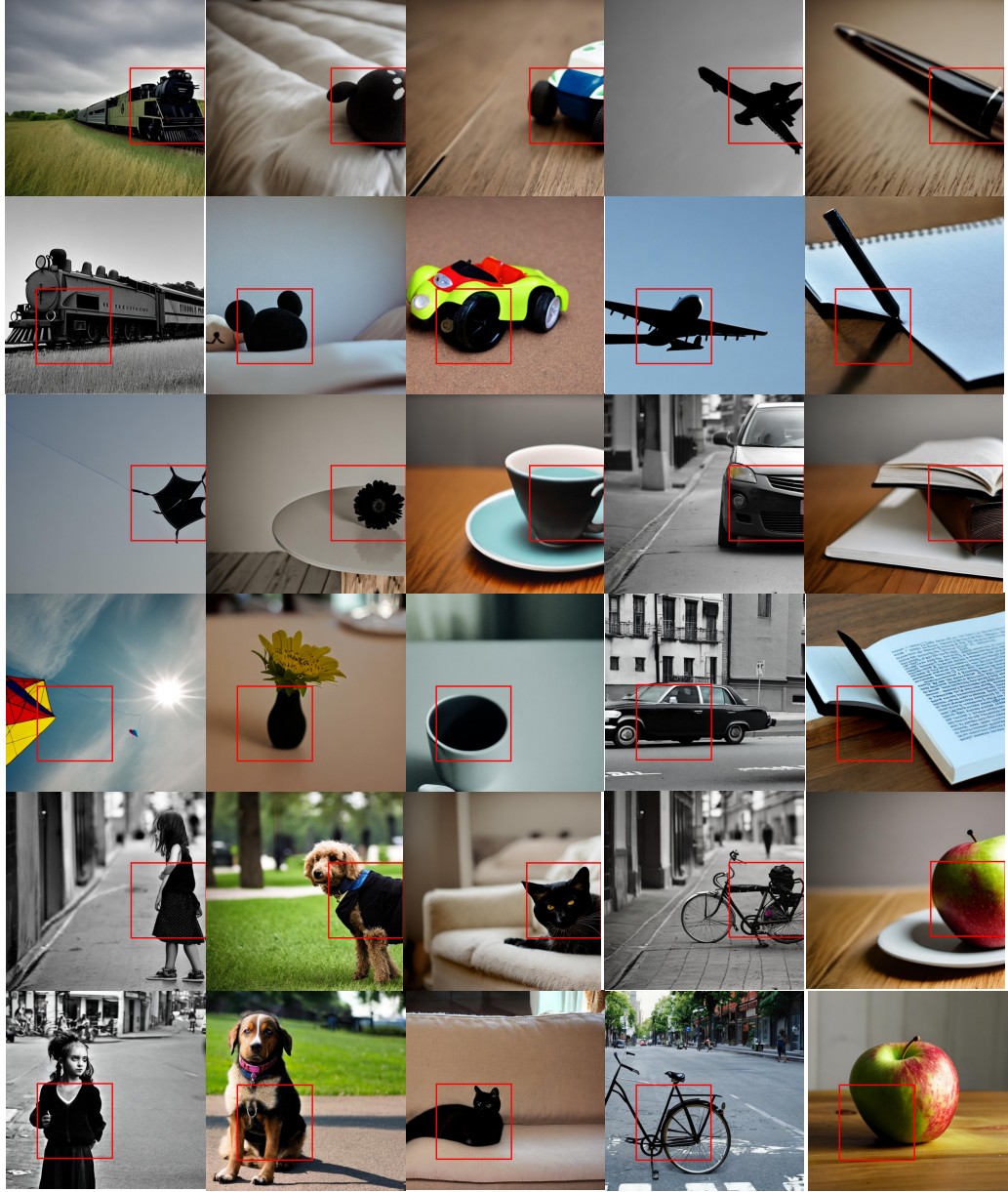

Figure 16: Failure analysis when the prompt aligns with the position of the trigger patch. In this case, there is a strong trigger patch located on the right edge of the image, while the prompt is "a stop sign on the right". However, the tree and the stop sign often appear together., leading to occasional instances where the tree occupies the intended position of the stop sign.

