# OpenReview forum: "The Crystal Ball Hypothesis in diffusion models: Anticipating object positions from initial noise"
_ICLR.cc/2025/Conference — ICLR 2025 Poster_

### Official Review · Reviewer_9YUS · 2024-10-31

**Soundness:** 3
**Presentation:** 3
**Contribution:** 3
**Rating:** 6
**Confidence:** 4

**Summary:**

This paper explores the influence of initial noise in the text-to-image generation tasks. They identify specific regions within the initial noise termed as trigger patches, which contribute to the positional bias in the generated images. In order to automatically discover the trigger patches, the authors train a detector. Next, the authors explore several special and interesting attributes of trigger patches, and discover the differences between the random patch within the initial noise and trigger patch. At last, based on the discoveries, the authors design two speical applications for trigger patch.

**Strengths:**

Thanks for the authors` efforts. I find this paper very interesting, and the experimental section is quite comprehensive. It has provided me with many insights and considerations.

1. The paper is well-organized, and the presentation of this paper is very good.

2. The motivation of this paper is clear, and it first explores the influence of positional bias in the initial noise. I think the experiments are quite comprehensive. I think this paper is valuable.

3. Some experiment results are very attractive, and they have sparked a lot of my thoughts and reflections.

**Weaknesses:**

Although, from my perspective, this paper is good, I think it still has several drawbacks:

Presentation:

1. In line 231, "As shown in the Figure", missing reference to the Figure. In line 293, missing (2) before "Can noises with multiple ...". In line 417, ":" is missing after "Control", and the first word after ":"， should be lowercase.  In line 1070, "Please refer to the Fig F.2 for visualized results.", I can not find the Fig.F2.

2. In Figure 13, the prompt is not consistent with the provided images.

3. All intuitive images in the main paper only contain five concepts. Can the authors provide more visualization results?

Methods and Experiments:

1. For the evaluation metrics, the authors only utilize the CLIP. I think it is crucial to utilize some human preference models like PickScore, HPSv2 to evaluate the quality of the generated images.

2. The dataset constructed by the authors only contains five single concepts, consisting of 500K samples. How do the authors ensure that whether the trigger patches exist beyond these five prompts, and the detector trained in this manner can recognize other concepts? Besides, I think the construction of the dataset is resource-consuming.

3. The authors claim that low entropy ISR will be higher; however, Table 6 shows that certain mid-entropy ISRs are also quite high. How should the authors explain this phenomenon?

4. All of the main results are conducted only in Stable Diffusion v2. Can the authors provide the existence of the trigger patches in other diffusion models?

5. Experiment results in Table 5 reveal “Random” can surpass "Initno" and "Attend-and-Excite". This phenomenon is very strange because as far as I know, both of them optimize the noise during the reverse process. They should be higher than "Random"?

6. The experiment results in Generalization part raise my concerns. From the results presented in Table 8, it seems that trigger patch injection is only effective with deterministic samplers, showing no impact on stochastic samplers. The authors also default to using a deterministic sampler. Besides, based on the results in Table 10, why is the gap reduction so pronounced across other models? If the initial noise inherently carries positional bias, then its performance across different diffusion models should be similar. However, the experimental results seem to contradict the notion that the trigger patch is universal.

**Questions:**

See the weaknesses.

---

> ### Comment · Reviewer_9YUS · 2024-11-24
>
> Wait for the authors to discuss with the reviewers.

---

> ### Author Response · Authors · 2024-11-24
> **Official Comment by Authors Part One**
>
> ### **Question about the presentation**
> Thank you for pointing it out! We have addressed the issues and made the necessary corrections. The revised version has been uploaded. More visualized results are added in Fig 15. Let us know if there are any further concerns or suggestions!
>
> ### **Question 1: Lack of human preference models.**
> Thank you for your advice! We utilized the **PickScore** model as suggested and compared the results of these methods against the control baseline. The score reflects the model's preference for the images generated by each method relative to the control baseline. The averaged results are presented as follows.
>
> | Method           | Structure Diffusion  | Composable Diffusion    | Attend-and-Excite | Divide-and-Bind | Ours |
> |------|---|------|--|------|--|
> | Score       | 0.5274 | 0.5256 | 0.5136 | 0.5336 | **0.5570** |
>
> Results show that our method surpasses all the baselines.
>
> ### **Question 2: Dataset limitation**
> Thank you for pointing it out! Our detector can certainly generalize to other objects. The key is that the trigger patches are **universal** across objects: **a trigger patch can be used to generate for any object.** As shown in the first row of Fig 1 and Fig 15, with different prompts, the same noise with a trigger patch at the right edge can generate different objects at the location, like
> ```
> 'apple', 'bike', 'book', 'car', 'cat', 'cup', 'dog', 'flower', 'girl', 'kite', 'pen', 'plane', 'toy', 'train', 'zebra'
> ```
> The universal property arises because these trigger patches are outliers within Gaussian noise, following distinct distributions as confirmed through two-sample tests (Section 4) **This indicates that the formation of trigger patches is driven by their unique statistical properties, rather than being tied to specific prompts or models.** Consequently, our detector identifies these outliers within the model, independent of object categories. This explains its strong generalizability to other objects.
>
> Meanwhile, Sec 5.1 shows the empirical evidence. **Here, we evaluate the detector in open-vocabulary scenarios, where generated objects are not limited to the original 5 classes** (see Appendix F.1 and Fig. 11 for details on prompts and visualized results). More specifically, in Section 5.1, we tested the detector on the following objects:
> ```
> 'person', 'bicycle', 'car', 'motorcycle', 'airplane', 'truck', 'fire_hydrant', 'bench', 'dog', 'handbag', 'frisbee', 'sports_ball', 'kite', 'baseball_glove', 'skateboard', 'surfboard', 'bottle', 'cup', 'knife', 'bowl', 'sandwich', 'orange', 'carrot', 'potted_plant', 'bed', 'dining_table', 'tv', 'laptop', 'mouse', 'keyboard'
> ```
> The strong results presented in Table 5 validate the generalizability of the detector.
>
> ### **Question 3: Conflicts in Fig 6 that Lower entropy contributes to higher ISR**
> It's a very good question! The reason is that the trigger entropy metric used in the paper cannot adequately model cases where **a noise contains multiple trigger patches.**
>
> For instance, in Figure 5, there are clearly two distinct trigger patches in the noise, located on the left and right sides of the image. However, since trigger entropy is computed for each noise as a whole (rather than for each trigger patch), it aggregates bounding boxes belonging to different patches into a single estimation. This leads to an incorrect assessment of the effectiveness of individual trigger patches. The trigger entropy of a noise containing two strong trigger patches might appear higher than that of a noise containing only one weaker trigger patch. This happens because bounding boxes from different trigger patches are viewed as part of the same patch, which distorts the evaluation.
>
> To address this issue, we propose analyzing the statistics of bounding boxes within a noise to directly extract the positions of individual trigger patches. By clustering bounding boxes in Figure 5, we can compute:
> - **Within-Cluster Variance (WCV):** This measures how tightly bounding boxes within a cluster (representing a trigger patch) are grouped.
> - **Between-Cluster Variance (BCV):** This measures the separation between clusters, where each cluster represents a distinct trigger patch.
>
> By treating each cluster as a representation of a trigger patch, the WCV of a cluster could serve as an enhanced trigger entropy, offering a better way to handle multiple trigger patches in a single noise.
>
> Additionally, **our detector naturally overcomes these issues by generating multiple bounding boxes for each noise.** Each bounding box is assigned a confidence score, which can be directly interpreted as the effectiveness of the trigger patch. This ensures that the detector evaluates each trigger patch independently, even in cases where multiple patches exist within a single noise sample. This approach avoids the aggregation problems of the original trigger entropy metric and provides a more accurate evaluation of trigger patch effectiveness.

---

> ### Author Response · Authors · 2024-11-24
> **Official Comment by Authors Part Two**
>
> ### **Question 4: Can trigger patch generalize to other models except for stable diffusion v2.**
> Thank you for your valuable advice! Through exhaustive experiments, we find out that our trigger patches generalize to Stable diffusion v1, Stable XL, and **DiT-based** Stable Diffusion v3[1].
>
> We verified it by following the protocol of the **Trigger Injection experiment** in Appendix C. Specifically, we injected the trigger patches into different random noise samples and used the blended noise to generate images. We then determined the success of the injection by checking whether the targeted region generated the intended object. A successful injection is defined as a detected bounding box occupying more than 75 percent of the trigger patch region. The Injection Success Rate (ISR) is calculated as the ratio of successful injections to the total number of cases, with **a higher ISR indicating a more obvious trigger patch phenomenon**.
> Following the steps outlined in Question 1,  we chose:
> 1. **Stable Diffusion XL**: We utilize the API from hugging face and diffusers.
> 2. **Finetuned models:** We used the off-the-shelf `oraul/finetuned_stable-diffusion-v1-4_FFHQ_smaller_ep_2` model from the Hugging Face Hub. It is fine-tuned from stable diffusion 1.4
> 3. **Stable Diffusion v3**: Note that all the previously discussed models are convolution-based, whereas Stable Diffusion v3 adopts a **DiT** backbone. To ensure compatibility, we modified the shape of the input latent to produce output images of the same size as those generated by other Stable Diffusion models. For implementation, we utilized the code from Hugging Face, using the model ID `stabilityai/stable-diffusion-3-medium-diffusers`. This adaptation allowed us to integrate the unique architecture of the DiT backbone while maintaining consistency in output dimensions for fair comparisons.
> For each configuration, we also conducted a resampling baseline, where Gaussian noise was resampled within the target patch while maintaining the same mean and variance. **A significant gap between the experimental group and the baseline indicates the effectiveness of the trigger patch in those settings.** The results are as follows:
>
> The results are as follows:
>
> | Models            | SDXL    | Oraul     | Stable Diffusion v3|
> |------|---|------|
> | Resampling Baseline       | 0.355   | 0.330   | 0.219 |
> | Experimental Group        | 0.500   | 0.525   |  0.441 |
> | Gap                       | 0.145   | 0.195   |  0.222 |
>
> The significant gap verifies that our trigger patch can generalize well to other diffusion models.
>
> [1] Esser, P., Kulal, S., Blattmann, A., Entezari, R., Müller, J., Saini, H., ... & Rombach, R. (2024, March). Scaling rectified flow transformers for high-resolution image synthesis. In Forty-first International Conference on Machine Learning.
>
>
> ### **Question 5: Why can “Random” surpass “Initno” and “Attend-and-Excite” in Table 5?**
> Thank you for your insightful comment! However, there might be a slight misunderstanding about the **Random baseline** and its role in the diversity task.
>
> The goal of the diversity task is to increase the positional diversity of the generated images. Ideally, **each pixel in the image should have an equal probability of generating an object**. To simulate this ideal scenario, the designed Random baseline does not generate images but instead:
> - **Random Bounding Box Selection:** It selects random bounding boxes of size 24×24 from the initial noise.
> - **Estimation of Ideal Diversity:** These randomly selected bounding boxes are treated as the object boxes for the **"ideally generated"** images.
> - **Entropy Computation:** The entropy is computed directly on these bounding boxes, providing a coarse **upper bound** for the ideal diversity case, which is the target of our diversity task.
>
> The results in Table 5 show that our method achieves similar entropy to the Random Baseline, which means the generated positions are very diverse. Approaches like Initno and Attend-and-Excite are not specifically designed for this task. These methods rely on accessing the generation process and modifying the feature maps case by case during image generation. Such modifications may inadvertently harm their performance when applied to this specific diversity-oriented application. So, these methods fall significantly short of the Random baseline.

---

> ### Author Response · Authors · 2024-11-24
> **Official Comment by Authors Part Three**
>
> ### **Question 6: Concerns on the generalization ability.**
> Thank you for your valuable insight!
>
> **About the sampler**: Our trigger patch does indeed perform worse with deterministic samplers compared to stochastic ones. To conduct a comprehensive quantitative analysis, we selected five stochastic samplers from the Diffusers library, as follows: DPMSolverSDEScheduler, KDPM2AncestralDiscreteScheduler, EulerAncestralDiscreteScheduler, TCDScheduler, and DDPMScheduler. We also selected four deterministic samplers discussed in Appendix C. We computed the ISR gap following the settings outlined in Q4. The results are summarized as follows:
> The higher the GAP is, the more effective the trigger patch is.
>
> | Deterministic Schedulers              | PNDM   | LMSD     | EulerDiscrete | HeunDiscrete | DDIM |
> |----|------|-----|-----|------|--------|
> | Resampling Baseline     | 0.105  | 0.095   | 0.115           | 0.110                 | 0.145 |
> | Experimental Group       | 0.660  | 0.620   | 0.620             | 0.610          |  0.630 |
> | Gap                                | 0.555  | 0.525   | 0.505             | 0.500          | 0.485 |
>
> | Stochastic Schedulers              | DPM   | KDPM    | EulerAncestral    | TCDS | DDPM|
> |----|------|-----|-----|------|--------|
> | Resampling Baseline      | 0.180  | 0.240   | 0.180  | 0.105           | 0.145                 |
> | Experimental Group       | 0.690  | 0.405  | 0.165  | 0.650                 | 0.155                 |
> | Gap                                  | 0.510  | 0.165  | -0.015  | 0.545                 | 0.010                 |
>
> As observed from the table, our method underperforms with certain stochastic samplers. The reason lies in the nature of stochastic samplers, which introduce noise to the latent variables at each diffusion step. If the added noise is large enough, it can overwrite the initial noise to some extent, diminishing the influence of our trigger patch. **Conversely, if the introduced noise is small, our trigger patch is still able to exert a strong effect.**
> To test this hypothesis, we conducted a series of experiments using the DDIM sampler from the diffusers, which includes a parameter `eta` to control the strength of the introduced noise Setting `eta = 0.0` makes the sampler behave like a deterministic one. The suggested value for eta is 0.1~0.3. We test different etas and the results are as follows:
>
> | eta              | 0.0   | 0.01    | 0.05    | 0.1 | 0.2 | 0.5 |
> |----|------|-----|-----|------|--------|-----|
> | Resampling Baseline      |  0.145 |  0.140  | 0.180  | 0.165    | 0.145                 | 0.160 |
> | Experimental Group       | 0.630 | 0.620  | 0.635  | 0.640       | 0.555                 | 0.355 |
> | Gap                                  |  0.485 | 0.480  | 0.455  | 0.475     | 0.410                | 0.195 |
>
> As we can see from the table, there is a clear **negative correlation between the eta parameter and the ISR GAP**. This demonstrates that the more noise the sampler introduces, the less effective the trigger patch becomes.
>
> **About the models **: Our trigger patch can transfer to other models in a **black-box manner**, but it may experience a loss in effectiveness. This is likely because different models use varying architectures and training datasets, which can introduce slight biases that affect the performance of the trigger patches. In fact, since each model has different mechanisms for generation, it is not expected that random patches for a particular model can fully transfer to another very different model.
>
> To address this, we can potentially improve the generalizability and create a comprehensive dataset using multiple models trained on diverse architectures and datasets. This ensembling-based training approach is expected to mitigate biases introduced by model-specific variations, thereby significantly improving the transferability of the trigger patch across different models, including black-box scenarios. We leave this exploration for future work.

---

> > ### Comment · Reviewer_9YUS · 2024-11-25
> >
> > Thanks for the authors' efforts. Good Luck!

---

### Official Review · Reviewer_Cp39 · 2024-11-03

**Soundness:** 3
**Presentation:** 3
**Contribution:** 3
**Rating:** 5
**Confidence:** 3

**Summary:**

The work finds out that there are some locations in the image frame where the diffusion often gets biased. These locations are marked as trigger patches. An object detector is trained to detect these patches before the sampling process. After that, rejection sampling is used to reject samples with trigger patches to obtain the better generative image diversity.

**Strengths:**

Strength:
1. The paper provides an interesting point of view on diffusion sampling
2. The new sampling methods help to increase the diversity

**Weaknesses:**

Weaknesses:
1. According to Figure 1 and Figure 3, the trigger patches are always in the right part of the image and only applicable to some objects. The whole paper also only focuses on some round objects like balls and tennis balls. This results in the question of generalization of the work. Whether or not this phenomenon will happen across datasets or with different objects?
2. The experimental results do not show the quantitative values to measure the output quality. There could be a trade-off between diversity and quality according to the new rejection sampling.
3. The current diversity result is not standard in generative measures such as FID/Recall.

**Questions:**

1. Please test on wider range of objects as well as datasets or find the trigger patches on different locations of the dataset instead of the right edge of the photo.
2. Please provide the qualitative values for image quality and observe the trade-off between diversity and image quality from the proposed rejection sampling method.
3. Please provide more comprehensive evaluations on diversity such as FID/Recall.

---

> ### Author Response · Authors · 2024-11-24
> **Official Comment by Authors Part One**
>
> ### **Question 1: Generalizability of trigger patches to different objects and positions**
> Thank you for the question. However, there seems to be a slight misunderstanding of our paper. Our trigger patches **can indeed generalize to various locations and objects**.
>
> **Positions of the trigger patches:**
>
> Note that Fig 1 only contains one noise with trigger patch on the right edge (first row) and all the images in the first row are generated from this noise. The same noise is used to visualize results in Fig 3. And this is just one particular example of a trigger patch. In fact, during the analysis of 20,000 noise samples, we encountered trigger patches located at **various positions** across the images.
> In Figure 8, we illustrated four cases where trigger patches were located at the **left**, **middle**, and **right** parts of the image. To further quantify the diversity of trigger patch positions, we analyzed the 1,000 noise samples with the **lowest trigger entropy** from the 20,000 noise samples in the dataset. We categorized the center points of the trigger patches into four regions. And the results are as follows:
>
> | Region           | Upper-left   | Upper-right    | Lower-left | Lower-right |
> |------|---|------|--|---|
> | Counts       | 264   | 198   | 294 | 244 |
>
> The results show that the location of the trigger patches is **balanced**, demonstrating that the trigger patches generalize well across different regions of the image.
>
> **Generalization of the trigger patches**:
>
> Although our detector is trained on a 5-class dataset, the trigger patches demonstrate clear generalization ability to other categories. The reason is that trigger patches are **universal** to different objects. The detector trained on 5 objects can generalize to other objects.
> In Section 5.1, we evaluate the detector in open-vocabulary scenarios, where generated objects are not limited to the original 5 classes (see Appendix F.1 and Fig. 11 for details on prompts and visualized results). More specifically, in Section 5.1, we have tested the detector on the following objects:
> ```
> 'person', 'bicycle', 'car', 'motorcycle', 'airplane', 'truck', 'fire_hydrant', 'bench', 'dog', 'handbag', 'frisbee', 'sports_ball', 'kite', 'baseball_glove', 'skateboard', 'surfboard', 'bottle', 'cup', 'knife', 'bowl', 'sandwich', 'orange', 'carrot', 'potted_plant', 'bed', 'dining_table', 'tv', 'laptop', 'mouse', 'keyboard'
> ```
> Good results in Table 5 verify its strong generalization capability to other objects.
>
> Additionally, in Appendix G, we test the trigger patches in multiple object generation settings. The objects we use are far beyond the original 5-classes. We list them here:
> ```
> 'monkey', 'bear', 'dog', 'frog', 'rabbit', 'horse', 'bird', 'elephant', 'turtle', 'cat', 'lion', 'mouse', 'car', 'backpack', 'suitcase', 'bow', 'bowl', 'clock', 'apple', 'crown', 'bench', 'chair', 'balloon' and 'glasses'.
> ```
> As shown in Table 11, our method outperforms the baselines, demonstrating its strong generalization capability to other objects.
>
> Finally, to provide additional visual examples, we present more illustrations similar to Fig. 1 in Fig. 15. This time, we include a variety of objects with diverse shapes and sizes.  The object categories are as follows:
> ```
> 'apple', 'bike', 'book', 'car', 'cat', 'cup', 'dog', 'flower', 'girl', 'kite', 'pen', 'plane', 'toy', 'train', 'zebra'
> ```
> Please refer to Fig 15 for more details.

---

> ### Author Response · Authors · 2024-11-24
> **Official Comment by Authors Part Two**
>
> ### **Question 2&3: Trade-off between image quality and diversity**
> **1. No obvious trade-off between diversity and quality is observed.**
>
> To gain a more quantitative analysis of the **image realism**, we computed the **Fréchet Inception Distance (FID)** of the generated images.  We used the validation set of ImageNet as the ground truth dataset. We believe it provides a reliable measure of how realistic our generated images appear.
>
> The FID results for Tab 5 in the location diversity application.
> | Methods      | Control | Initno | Attend | Refocusing | Structured | Ours   |
> |--------------|---------|--------|--------|------------|------------|--------|
> | **Entropy**  | 135.97  | 139.89 | 145.16 | 102.097    | 133.92   | 171.84 |
> | **FID**  | 112.76  | 115.61 | 113.51 | 117.08    | 119.62     | 113.59 |
>
> As we can see from the table, **no significant trade-off is observed.**  While it is slightly surpassed by Attend, it offers substantial advantages in terms of simplicity and efficiency: it requires no prior knowledge of the model or prompts and does not rely on predefined enhanced token IDs, as used by Attend. Meanwhile, it is also faster than Attend, as it doesn’t need to do optimization during generation.
>
> **2. Is FID reasonable for the diversity task?**
>
> While FID is commonly used to evaluate image realism, using it as a diversity metric might be unreasonable for several reasons. Objects in the ImageNet training data are usually located at the center of the image, creating a center bias. In contrast, our objective is to have objects disperse uniformly across the image. This conflict makes using a center-biased dataset like ImageNet as the target distribution unsuitable for evaluating the position diversity of our generated images. **So to perform a fair evaluation of our method’s image quality and position diversity, we need a dataset with diverse object positions.** This dataset would avoid the biases inherent in ImageNet and provide a more accurate benchmark for tasks requiring position diversity.
>
> However, our metric is specifically designed to directly assess position diversity. By calculating the variance of the position coordinates of the bounding boxes for the detected objects, it provides a clear measure of positional spread. **Lower variance indicates a higher concentration of objects, which corresponds to poor position diversity.** Please refer to Fig 3 for more details. In fact, the trigger entropy in Fig 3 is the metric we used in Sec 5.1.

---

> > ### Comment · Reviewer_Cp39 · 2024-11-27
> > **Further questions**
> >
> > May I check with the author about how you obtained the pre-trained diffusion model? What is the setting for dataset/image size for the table in the rebuttals? From my understanding, the diffusion models all achieve very low scores on most of the resolutions on ImageNet, around 1 - 30 FID (even with 10000 samples). But your results show around 100FID, this seems to be either very noisy images or very less diverse.
> >
> > I don't agree with the authors in the phrase, "While FID is commonly used to evaluate image realism, using it as a diversity metric might be unreasonable for several reasons". FID measures the distance between mean and variance between representative information of the generated dataset and target datasets. If the image has low diversity it will achieve very high FID due to large variance discrepancy. In some works like [1], section 2, it points out that FID measures intra-class mode dropping as a measure for diversity.
> >
> > If the authors don't agree with FID, there is another measure for it, which is the Recall value from [2].
> >
> > [1] Lucic, Mario, et al. "Are gans created equal? a large-scale study." Advances in neural information processing systems 31 (2018).
> > [2] Kynkäänniemi, Tuomas, et al. "Improved precision and recall metric for assessing generative models." Advances in neural information processing systems 32 (2019).

---

### Official Review · Reviewer_zuoZ · 2024-11-03

**Soundness:** 3
**Presentation:** 3
**Contribution:** 3
**Rating:** 6
**Confidence:** 4

**Summary:**

This article introduces the intriguing concept of trigger patch noise. The authors first demonstrate the existence of trigger patch noise, then propose a method using neural networks to detect this noise in initial noise. They experimentally verify the unique characteristics of trigger patch noise and explore its potential applications, such as controlling object generation position by embedding trigger patch noise into the initial noise.

**Strengths:**

1.The insight is interesting.

2.The authors not only demonstrated the existence of trigger patch noise through experiments but also provided a method for its detection. Additionally, the authors thoroughly explored the effects of multiple trigger patch noises, examined the relationship between trigger patch noise and prompt guidance, and investigated the generalization capabilities of trigger patch noise.

3.I believe that the proposed concept of trigger patch noise has the potential to advance research and development in the field of controllable generation.

**Weaknesses:**

1.The experiment was conducted with only five object categories. Can trigger patch noise be generalized to a broader range of object categories? This could provide insights into its robustness across diverse applications.

2.Does trigger patch noise commonly exist within any randomly sampled Gaussian noise? Additionally, if multiple trigger patch noises are present in a single Gaussian noise, what criteria should be used to select the most effective trigger patch noise?

3.In the training process of the trigger patch noise detector, how do you define the ground truth for trigger patch noise? Providing a detailed algorithm or clarification of this aspect would be valuable. Furthermore, does every one of the 17,000 noises in the training set contain trigger patch noise?

4.There seems to be an impact on image realism when inserting trigger patch noise. How significant is this effect, and are there any methods to mitigate it? (The BLIP-Text results in Table 11)

5.The detection accuracy of trigger patch noise reported in the paper appears somewhat limited. Are there plans or suggestions for enhancing detection performance?

6.What would be the effect of inserting multiple (e.g., three) trigger patch noises? Additionally, when aiming to control the size of the generated object, does resizing the trigger patch noise (e.g., scaling) still yield the intended effect?

7.Can the trigger patch noise work for DiT models?

**Questions:**

See weakness.

---

> ### Author Response · Authors · 2024-11-24
> **Official Comment by Authors Part One**
>
> ### **Question 1: Broader range object categories**
> Although our detector is trained on a 5-class dataset, the trigger patches demonstrate clear generalization ability to other categories. The reason is that trigger patches are **universal** to different objects. The detector trained on 5 objects can generalize to other objects.
> In Section 5.1, we evaluate the detector in open-vocabulary scenarios, where generated objects are not limited to the original 5 classes (see Appendix F.1 and Fig. 11 for details on prompts and visualized results). More specifically, in Section 5.1, we have tested the detector on the following objects:
> ```
> 'person', 'bicycle', 'car', 'motorcycle', 'airplane', 'truck', 'fire_hydrant', 'bench', 'dog', 'handbag', 'frisbee', 'sports_ball', 'kite', 'baseball_glove', 'skateboard', 'surfboard', 'bottle', 'cup', 'knife', 'bowl', 'sandwich', 'orange', 'carrot', 'potted_plant', 'bed', 'dining_table', 'tv', 'laptop', 'mouse', 'keyboard'
> ```
> The good results in Table 5 verify its strong generalization capability with other objects.
>
> Additionally, in Appendix G, we test the trigger patches in multiple object generation settings. The objects we use are far beyond the original 5-classes. We list them here:
> ```
> 'monkey', 'bear', 'dog', 'frog', 'rabbit', 'horse', 'bird', 'elephant', 'turtle', 'cat', 'lion', 'mouse', 'car', 'backpack', 'suitcase', 'bow', 'bowl', 'clock', 'apple', 'crown', 'bench', 'chair', 'balloon' and 'glasses'.
> ```
> As shown in Table 11, our method outperforms the baselines, demonstrating its strong generalization capability to other objects.
>
> Finally, to provide additional visual examples, we present more illustrations similar to Fig. 1 in Fig. 15. This time, we include a variety of objects with diverse shapes and sizes.  The object categories are as follows:
> ```
> 'apple', 'bike', 'book', 'car', 'cat', 'cup', 'dog', 'flower', 'girl', 'kite', 'pen', 'plane', 'toy', 'train', 'zebra'
> ```
> Please refer to Fig 15 for more details.
>
> ### **Question 2: Prevalence and Selection Criteria for Trigger Patches**
> **The trigger patch is relatively common in randomly sampled Gaussian noise.** In Fig. 4, we present histograms of the trigger entropy for 20,000 noise samples from our dataset. 57.61% of noises have trigger entropy less than 0.01 of the image size, signifying that for over half of the noises, the variance of the generated box coordinates is under 10% of the image size. This suggests that **in over half of the noises, the generated objects are confined to a bounding box of size 0.1x the image dimensions.**
>
> **Selection criteria when multiple trigger patches are present:** The **confidence score** of the detector, as introduced in Section 3.1, serves as a reliable indicator of the effectiveness of trigger patches. When multiple trigger patches are present in an image, the detector can be utilized to identify and rank them based on their confidence scores. As described in Section 5, we consider a patch to be an effective trigger only if its detector confidence score exceeds 0.7.
>
> ### **Question 3: Training data labeling and data statistics labeling.**
>
> **Ground truth labeling**
> We first generate images using the noise and apply a pre-trained object detector from MMDetection to identify the object bounding boxes. The bounding boxes are then divided by 8 to transform them from the 512x512 image space to the 64x64 latent space. For each image, only the bounding box with the highest score and the correct object label is retained. Then the bounding boxes are viewed as the ground truth labels.
>
> **Does every noise contain a trigger patch?**
> All the noise has trigger patches; however, their effectiveness varies. Not all noise patterns contain a (strong) trigger patch.
> For instance, in Fig. 5, we present a histogram of the trigger entropy for 20,000 noise samples. Among these, over 100 noises exhibit a variance exceeding 150.
> Referring back to the **Random Baseline** in Sec. 5.1, the objective of the diversity task is to enhance the positional diversity of generated images. Ideally, every pixel in the image should have an equal probability of generating an object. To approximate this ideal scenario, the Random Baseline selects random bounding boxes of size 24×24 from the initial noise and computes the entropy for these bounding boxes, providing a coarse upper bound for the ideal diversity case—the target for our diversity task. **Notably, the trigger entropy of the tested methods approaches 150, which is very close to the Random Baseline's entropy of 170.** This indicates that there are no strong trigger patches present in the analyzed noise patches.

---

> ### Author Response · Authors · 2024-11-24
> **Official Comment by Authors Part Two**
>
> ### **Question 4: Impact on the image realism.**
> **Our method achieves comparable realism quality.** To evaluate the realism of the generated images, we compute the Fréchet Inception Distance (FID), using the ImageNet validation set as the ground truth. This metric effectively measures how realistic the generated images are.
>
> The FID results for Tab 5 in the location diversity application are as follows:
> | Methods    | Control   | Initno    | Attend    | Refocusing | Structured | Ours      |
> |------------|-----------|-----------|-----------|------------|------------|-----------|
> | **Entropy**| 135.97    | 139.89    | 145.16    | 102.10     | 133.92     | **171.84**|
> | **FID**    | 112.76    | 115.61    | 113.51    | 117.08     | 119.62     | **113.59**|
>
> As we can see from the table, our method demonstrates **competitive FID performance** compared to other approaches. While it is slightly surpassed by Attend, it offers substantial advantages in terms of simplicity and efficiency: **it requires no prior knowledge of the model or prompts and does not rely on predefined enhanced token IDs, as used by Attend.** Meanwhile, it is also faster than Attend, as it doesn’t need to do optimization during generation.
>
> ### **Question 5:  Limited performance of the detector**
>
> Note that Faster R-CNN [1] achieves an mAP50 of 0.596 on the COCO dataset. Our accuracy, while lower, is comparable for detecting X (X = a specific class) in COCO, achieving an mAP50 of 0.333. The gap, while notable, is not excessively wide.
> However, there are underlying reasons for the limited performance, and the potential remedies are as follows:
>
> **1. Drawbacks in Ground Truth Labeling:**
>
> As shown in Fig. 4 and Fig. 5, random sampled noise can contain multiple or no trigger patches, which was not considered when generating ground truth labels. Specifically:
> - **No trigger patch:** When noise contains no trigger patch, it results in unreasonable data points.
> - **Multiple trigger patches:** Bounding boxes may concentrate on one patch while ignoring others, leading to unbalanced data points.
> - **Noisy labels:** Each noise can generate up to 25 bounding boxes, many noisy or repetitive.
>
> **Remedy:** Analyze the statistics of bounding boxes for each noise to extract trigger patch positions. By clustering bounding boxes (similar to Fig. 5), we can compute Within-Cluster Variance (WCV) and Between-Cluster Variance (BCV). If WCV exceeds a threshold, the noise lacks a trigger patch and should be filtered out. Each cluster can represent a trigger patch, and the averaged bounding box for each cluster can serve as the ground truth label.
>
> ---
>
> **2. Challenges with Advanced Techniques:**
>
> Advanced techniques commonly used in object detection, like random cropping in data augmentation, reduced performance for our task due to the unique nature of our input data. We found out that simplified augmentation techniques were more effective, retaining statistical properties. Finally, we adopted only random horizontal flip and crop for data augmentation.
>
> **Remedy:** The theory presented in Section 4 suggests that these trigger patches are statistical outliers. So methods like **vertical flip, rotation, and channel permutation** should also be effective since they preserve the statistical properties. We leave it as future work.
>
> ---
>
> **3. Object Size Variance:**
> Object sizes in our dataset vary significantly. For instance, in a 512x512 image, the average area of a bear is 69,274 pixels, whereas for a sports ball, it is only 29,919 pixels. Real-world scenarios also involve significant object size variation.
> **Remedy:** Future training should include more objects of varied sizes to improve detector robustness in real-world applications.
>
> [1] Ren, Shaoqing. "Faster r-cnn: Towards real-time object detection with region proposal networks." arXiv preprint arXiv:1506.01497 (2015).

---

> ### Author Response · Authors · 2024-11-24
> **Official Comment by Authors Part Three**
>
> ### **Question 6.1:  Inserting multiple trigger patch noises**
> We found out that by strategically placing multiple trigger patches in specific regions, we can influence the **overall layout**, and finally minimize issues such as missing or fused objects, thereby **improving the overall image quality and diversity**. Please refer to Appendix G for more details, and here is the summary:
>
> Following the protocol outlined in Attend-and-Excite [1], we used the same prompts provided in their GitHub repository and generated 64 images for each prompt. Below are some examples of the prompts used:
> ```
>         "a lion and a yellow clock", "a bird and a black bowl", "a monkey and a red car", "a mouse with a bow", "a frog and a purple balloon",
> ```
> For our method, we place two separate trigger patches on the left and right sides of the image. In that way, we can avoid the two objects fusing into each other. In detail, the latent has a size of 64 × 64. We place the two trigger patches within the bounding boxes defined by the coordinates (0, 20, 24, 44) and (40, 64, 24, 44), where the four coordinates represent the top-left and bottom-right corners of each bounding box in the format (x1, x2, y1, y2). This ensures that the patches are placed in distinct regions of the image, **preventing any overlap or fusion of the objects associated with these triggers.** **For more visualized results, please refer to Figure 2 in the attached PDF.**
> We then evaluated the results using three metrics: CLIP Full Image-Text Similarity (CLIP-Full), CLIP Minimum Image-Text Similarity (CLIP-Min), and BLIP-CLIP Text-Text Similarity (BLIP-Text) as proposed in the paper. For all these metrics, **higher values indicate better performance**. The results are as follows:
>
> | Methods   | Stable Diffusion | Composable Diffusion[2] | Structure Diffusion[3] | Attend-and-Excite[1] | Divide-and-Bind[4] |  Ours       |
> |-----------|-------------------|------------------------|-----------------------|---------------------|------------------|-------------|
> | CLIP-Full | 0.341              | 0.348                  | 0.349                 | 0.352               | 0.349            | **0.360**   |
> | CLIP-Min  | 0.251              | 0.252                  | 0.246                 | 0.263               | 0.261        | **0.264**       |
> | BLIP-Text | 0.792              | 0.768                  | 0.780                 | **0.830**               | 0.822            | 0.826   |
>
> In both CLIP-Full and CLIP-Min metrics, our method achieved the best results, surpassing the strongest baselines that specifically designed for this task. by 0.008 and 0.001, respectively. Additionally, our method does **not require any intervention in the attention maps** during the generation process, which significantly reduces the computational resources needed and makes it model and prompt irrelevant.
>
> [1] Chefer, H., Alaluf, Y., Vinker, Y., Wolf, L., & Cohen-Or, D. (2023). Attend-and-excite: Attention-based semantic guidance for text-to-image diffusion models. ACM Transactions on Graphics (TOG), 42(4), 1-10.
>
> [2] Liu, N., Li, S., Du, Y., Torralba, A., & Tenenbaum, J. B. (2022, October). Compositional visual generation with composable diffusion models. In European Conference on Computer Vision (pp. 423-439). Cham: Springer Nature Switzerland.
>
> [3] Feng, W., He, X., Fu, T. J., Jampani, V., Akula, A., Narayana, P., ... & Wang, W. Y. (2022). Training-free structured diffusion guidance for compositional text-to-image synthesis. arXiv preprint arXiv:2212.05032.
>
> [4] Guo, X., Liu, J., Cui, M., Li, J., Yang, H., & Huang, D. (2024). Initno: Boosting text-to-image diffusion models via initial noise optimization. In Proceedings of the IEEE/CVF Conference on Computer Vision and Pattern Recognition (pp. 9380-9389).
>
> ### **Question 6.2: Can resizing the trigger patch noise control the size of the generated object?**
>
> Thank you for this interesting question. **We found that resizing the trigger patch indeed impacts the size of the generated object.** To validate this, we followed the injection experiment protocol described in L249 and Appendix C. Specifically, a trigger patch of size `24x24` was resized into four resolutions: `12x12`, `18x18`, `24x24`, and `30x30`. Using the Trigger Injection setup illustrated in Figure 6, the trigger patch was injected into randomly sampled noise, with the top-left corner fixed at `[10, 30]`.
>
> A pre-trained COCO detector was then employed to compute the **average area** of the resulting bounding boxes. The results are summarized in the table below:
>
> | **Size**   | **12x12** | **18x18** | **24x24** | **30x30** |
> |------------|-----------|-----------|-----------|-----------|
> | **Area**   | 694.1     | 1101.8    | 1481.99   | 1597.01   |
>
> The findings reveal a strong positive correlation between the size of the trigger patch and the size of the generated objects, confirming that the trigger patch size significantly influences the resulting object size.

---

> > ### Author Response · Authors · 2024-11-24
> > **Official Comment by Authors Part Four**
> >
> > ### **Question 7: Can these trigger patches generalize to DiT models?**
> > Thank you for your valuable advice! We found that our trigger patches generalize to DiT-based models.
> >
> > We verified it by following the protocol of the **Trigger Injection experiment** in Appendix C. Specifically, we injected the trigger patches into different random noise samples and used the blended noise to generate images. We then determined the success of the injection by checking whether the targeted region generated the intended object. A successful injection is defined as a detected bounding box occupying more than 75 percent of the trigger patch region. The Injection Success Rate (ISR) is calculated as the ratio of successful injections to the total number of cases, with **a higher ISR indicating a more obvious trigger patch phenomenon**.
> >
> > In addition to Table 10 where we test the ISR of stable diffusion XL, finetuned models, and Lora models, we adopt Stable diffusion v3[1] with a **DiT** backbone. We modify the shape of the input latent to make the output image the same size as the other stable diffusion models. We utilize the code from hugging face and the model ID is `stabilityai/stable-diffusion-3-medium-diffusers`. All other configurations remained consistent with those in Appendix C.
> >
> > For each configuration, we also conducted a resampling baseline, where Gaussian noise was resampled within the target patch while maintaining the same mean and variance. **A significant gap between the experimental group and the baseline indicates the effectiveness of the trigger patch in those settings.** The results are as follows:
> >
> > The results are as follows:
> >
> > | Models            | SDXL    | Oraul     | Water Color |  Crayons | Stable Diffusion 3|
> > |------|---|------|--|---|------|
> > | Resampling Baseline       | 0.355   | 0.330   | 0.265 | 0.305 | 0.219 |
> > | Experimental Group        | 0.500   | 0.525   |  0.565 |0.695 | 0.441 |
> > | Gap                       | 0.145   | 0.195   | 0.300 | 0.390 |  0.222 |
> >
> > The significant gap verifies the great effectiveness of our trigger patch in a DIT model.
> >
> > [1] Esser, P., Kulal, S., Blattmann, A., Entezari, R., Müller, J., Saini, H., ... & Rombach, R. (2024, March). Scaling rectified flow transformers for high-resolution image synthesis. In Forty-first International Conference on Machine Learning.

---

> ### Comment · Reviewer_zuoZ · 2024-11-26
>
> Based on the experiment results and limitations of the paper as well as the answers of rebuttal, we decide to keep my score.

---

> > ### Author Response · Authors · 2024-12-03
> >
> > Thank you for your reply!
> > ### **Question 1: Experiment configurations for FID.**
> > We load the pre-trained models from the Diffusers repository using the model ID `stabilityai/stable-diffusion-2-base`. For each baseline, we generate 10,000 images at a resolution of 512x512.
> > **The high FID score may be attributed to the prompts failing to cover all ImageNet classes.** To analyze the generated images, we use a COCO detector to identify the types of objects present in the generated images in Sec 5.1. Results show that only 30 out of the 80 classes are detected:
> > ```
> > 'person', 'bicycle', 'car', 'motorcycle', 'airplane', 'truck', 'fire_hydrant', 'bench', 'dog', 'handbag', 'frisbee', 'sports_ball', 'kite', 'baseball_glove', 'skateboard', 'surfboard', 'bottle', 'cup', 'knife', 'bowl', 'sandwich', 'orange', 'carrot', 'potted_plant', 'bed', 'dining_table', 'tv', 'laptop', 'mouse', 'keyboard'
> > ```
> > For certain indoor objects, such as "monitor" and "mouse," we observe that they have not been generated.
> >
> > ### **Question 2: Discussions about FID.**
> > We appologize for the confusion in the referred sentence: We agree that FID can measure object diversity (how diverse are the objects in the generated images). What we meant to say is that it cannot measure **positional diversity** properly, especially when the target distribution is biased.
> >
> > 1. **Center Bias exists in the target dataset: ImageNet**:
> >     - ImageNet, the dataset often used as a reference for FID calculation, exhibits a **center bias**, where objects are typically located in the center of the image. This bias affects the calculation of FID because the reference distribution does not represent a diverse spatial layout of objects.
> >     - When evaluating generative models for **position diversity**, using a center-biased dataset as the target distribution is inherently problematic. This is because the generative model's goal of dispersing objects across the image conflicts with the biased reference dataset. You argue that *low diversity in images would result in very high FID due to a large variance discrepancy.* However, this may not necessarily be the case. If the objects in both the biased ImageNet dataset and the generated images are similarly concentrated, **the component representing positional variance would also be equally low and close**, leading to a small discrepancy instead.
> >
> > 2. **FID Measures Overall Realism, Not Position Diversity**:
> >     - FID compares the overall feature distributions of generated and real images (based on the Inception network's intermediate features) but does **not** directly measure specific aspects like the spatial distribution of objects.
> >     - It may be a good measurement for **object diversity** as these high-level semantic contents referring to object categories are abundant in these features. However it may fail to measure positional diversity.
> >     - Previously, we refer to “realistic” mostly to the diversity in object diversity and their appearance. However, a model generating highly realistic but centrally biased images could score well on FID while failing to achieve position diversity.
> > 3. **Alternative Metrics for Position Diversity**:
> > The misfit of FID is also shared by recall value. For these reasons, we propose using **variance in bounding box coordinates** or **trigger entropy** as metrics to directly assess position diversity. These metrics are designed to capture the spatial spread of objects, which FID cannot measure.
> >
> > In conclusion, ImageNet exhibits positional bias but has no object bias. Therefore, using FID on ImageNet to evaluate the diversity of objects in generated images is acceptable, whereas relying on it to assess positional bias may be problematic.

---

### Official Review · Reviewer_Tsox · 2024-11-03

**Soundness:** 3
**Presentation:** 3
**Contribution:** 3
**Rating:** 8
**Confidence:** 3

**Summary:**

In the paper, the authors introduce the concept of "trigger patches," which are specific regions within the initial noise image that determine the positional information in object generation. They present a method for localizing these trigger patches, provide analyses to validate their correctness, and demonstrate two applications.

**Strengths:**

1. The concept of "trigger patches" in the diffusion generation process is both interesting and useful, as it enhances the controllability of the generation.
2. The experiments and analyses are thorough and comprehensive.
3. The writing is clear and intuitive.

**Weaknesses:**

1. In the evaluation of the second application (Sec 5.2), only "left" and "right" are considered. Is it possible to include more fine-grained positional information, such as "left-down"?
2. In the introduction, it is mentioned that "moving/removing trigger patches can achieve certain image editing effects." Including some image editing results in the experimental section would support this claim.
3. In cases where "trigger patches" and prompts are aligned, the positional generation accuracy is 63.5%. What factors contribute to the failures in the remaining cases? Can the authors provide a failure analysis?

**Questions:**

Please address the questions / concerns in the weaknesses section.

---

> ### Author Response · Authors · 2024-11-24
>
> ### **Question 1: Only "left" and "right" are considered.**
> Thank you for your suggestion. To address the question, we split the image into four regions: left-up, left-down, right-up, right-down, and generate images using prompts that incorporate these words, e.g., "a sports ball in the left-up corner". We then used a pre-trained COCO detector from MMDetection to verify whether the objects were correctly positioned according to the prompt.
>
> Supposing the x-axis coordinates of the left and the right edges of the bounding boxes are $x_1$ and $x_2$ respectively, and the image has a size of $512 \times 512$, we define the generated object in the left part of the image as:
> $$
> \frac{x_1 + x_2}{2} < \frac{512}{2}
> $$
> and the generated object in the down part of the image as:
> $$
> \frac{y_1 + y_2}{2} > \frac{512}{2}
> $$
>
> If *the generated object in the left part of the image* and *the generated object in the down part of the image* align with the prompt specifying "left-down,", we consider it a successful case.
>
> We calculate the **GSR** (Generated Success Rate) as the ratio of the number of successful cases to the total number of cases. The higher the GSR, the more effective the method.
>
> | Methods             | Control | Random | Attend | Attention Refocusing | Structured | Ours   |
> |---------------------|---------|--------|--------|-----------------------|------------|--------|
> | **GSR (%)**         |  27.2  | 29.0  | 32.1  | 53.6                 | 44.9      | **51.8** |
>
> Results above show that our method generalizes well in this setting.
>
> ### **Question 2: Image editing results**
> Fig 1 actually shows the image editing results.
> Imagine a user generating an image using the prompt: "A grizzly bear fishes in a rushing river." (the third prompt in Figure 1) and the resulting image places the bear at the right edge (top figure of column 3). If this is not desirable, we can move the trigger patch to the desired location, such as the bottom-left region. As a result, the bear will be generated in the intended position (bottom figure of column 3).
>
> To provide additional visual examples, we present more illustrations in Fig. 15. This time, we include a variety of objects with diverse shapes and sizes. The object categories are as follows:
> ```
> 'apple', 'bike', 'book', 'car', 'cat', 'cup', 'dog', 'flower', 'girl', 'kite', 'pen', 'plane', 'toy', 'train', 'zebra'
> ```
> Please refer to Fig 15 for more details.
>
> ### **Question 3: Failure analysis for aligned experiments**
> There are mainly three reasons for the failed cases for aligned experiments.
>
> Firstly, since our trigger patches are **universal**, any object can be generated at the designated position. Occasionally, irrelevant objects may occupy this position. For instance, given the prompt “a stop sign in the street” with the trigger patch located in the right corner, the position might instead be occupied by a tree. Please refer to the Fig 16 for the visualized results.
>
> Secondly, some classes we use have difficulty determining whether they are on the left or right. For example, for the class "fork," the model often generates a long fork that stretches across the image horizontally. Finally, in some cases, the objects are placed almost in the middle of the image and It is hard to judge if it is correct. These two cases will be categorized into “hard to judge”.
>
> We can observe that although “aligned” only has 63.5%, most of the rest samples belong to “hard to judge”, and there are very few samples belonging to “contradicted” and “duplicated”.  As seen from the bottom row in Table 3, when the initial noise has no strong trigger patch, the cases belonging to hard to judge can exceed 50%, showing the trigger patch plays an important role in the position determination. In fact, if we ignore the hard-to-judge cases, there are 83.4% cases belonging to “aligned” while only 16.6% cases belong to “contradicted” and “duplicated”.

---

### Meta-Review · Area_Chair_6t7k · 2024-12-17

**Metareview:**

This paper explores the influence of initial noise in the text-to-image generation task, and proposes the concept of triggle patches that play a key role in inducing object generation in the resulting images. All reviewers agree the insight of triggle patches are interesting, and the discovery of triggle patches helps to advance research in the field of controllable generation. The AC agrees with the reviewers and recommend accept.

**Additional Comments On Reviewer Discussion:**

This paper received mixed scores.

Most reviewers argue the generalization of the method. Such as the experiments are only conducted with only five object categories (Review zuoZ, Cp39, 9YUS), the transfer ability for DiT models (Review zuoZ, 9YUS), also Reviewer 9YUS suggest adding human preference models such as pickscore for evaluation.

After rebuttal, most questions are fixed. The author should pay attention that, as Reviewer Cp39 further questioned, the experimental setting in Table 5, and the rationality of using FID for quantitative evaluation, the authors should clarify this in the revised version.

---

### Decision · Program_Chairs · 2025-01-22

Accept (Poster)